# A glutamatergic DRN–VTA pathway modulates neuropathic pain and comorbid anhedonia-like behavior in mice

Xin-Yue Wang[1,6], Wen-Bin Jia[1,6], Xiang Xu[1], Rui Chen[1], Liang-Biao Wang[1], Xiao-Jing Su[1], Peng-Fei Xu[1], Xiao-Qing Liu[2], Jie Wen [3], Xiao-Yuan Song [4], Yuan-Yuan Liu [5], Zhi Zhang [4] ✉, Xin-Feng Liu [1] ✉ & Yan Zhang [1] ✉

Chronic pain causes both physical suffering and comorbid mental symptoms such as anhedonia. However, the neural circuits and molecular mechanisms underlying these maladaptive behaviors remain elusive. Here using a mouse model, we report a pathway from vesicular glutamate transporter 3 neurons in the dorsal raphe nucleus to dopamine neurons in the ventral tegmental area (VGluT3$^{DRN}$→DA$^{VTA}$) wherein population-level activity in response to innocuous mechanical stimuli and sucrose consumption is inhibited by chronic neuropathic pain. Mechanistically, neuropathic pain dampens VGluT3$^{DRN}$ → DA$^{VTA}$ glutamatergic transmission and DA$^{VTA}$ neural excitability. VGluT3$^{DRN}$ → DA$^{VTA}$ activation alleviates neuropathic pain and comorbid anhedonia-like behavior (CAB) by releasing glutamate, which subsequently promotes DA release in the nucleus accumbens medial shell (NAcMed) and produces analgesic and anti-anhedonia effects via D2 and D1 receptors, respectively. In addition, VGluT3$^{DRN}$ → DA$^{VTA}$ inhibition produces pain-like reflexive hypersensitivity and anhedonia-like behavior in intact mice. These findings reveal a crucial role for VGluT3$^{DRN}$ → DA$^{VTA}$ → D2/D1$^{NAcMed}$ pathway in establishing and modulating chronic pain and CAB.

Pain is a mixture of sensory/discriminative and motivational/affective experiences. The nervous system elegantly interprets pain and pain relief as aversive and rewarding processes, which trigger specific motivated behavioral responses. Acute pain results in adaptive escape/avoidance actions to prevent further tissue damage, and relief from acute pain is rewarding to facilitate learning of how to predict dangerous or rewarding situations in the future[1,2]. However, chronic pain, which affects nearly 20% of individuals worldwide[3], causes maladaptive physical suffering and mental symptoms, including comorbid

anhedonia, which may conversely reinforce the intensity and duration of pain[4]. The vicious cycle created by pain and its negative effect often results in chronic, refractory pain for which medical management is challenging, and an improved understanding of the mechanisms underpinning chronic pain is needed to develop novel therapeutic strategies for effectively treating pain without adverse side effects.

The mesolimbic dopamine system, which mainly comprises dopamine neurons in the ventral tegmental area (DA$^{VTA}$ neurons) projecting to the medial prefrontal cortex (mPFC) and nucleus

[1]Department of Neurology, The First Affiliated Hospital of USTC, Division of Life Sciences and Medicine, University of Science and Technology of China, 230001 Hefei, China. [2]School of Basic Medical Sciences, Division of Life Sciences and Medicine, University of Science and Technology of China, 230027 Hefei, China. [3]Department of Radiology, The First Affiliated Hospital of USTC, Division of Life Sciences and Medicine, University of Science and Technology of China, 230001 Hefei, China. [4]Hefei National Research Center for Physical Sciences at the Microscale, Division of Life Sciences and Medicine, University of Science and Technology of China, 230026 Hefei, China. [5]Somatosensation and Pain Unit, National Institute of Dental and Craniofacial Research (NIDCR), National Center for Complementary and Integrative Health (NCCIH), National Institutes of Health (NIH), Bethesda, MD, USA. [6]These authors contributed equally: Xin-Yue Wang, Wen-Bin Jia. ✉e-mail: zhizhang@ustc.edu.cn; xfliu2@ustc.edu.cn; yzhang19@ustc.edu.cn

accumbens (NAc), is believed to be the primary machinery for encoding motivation, reward, and aversion[5]. A number of human and animal studies have shown that DA release in the NAc is reduced in chronic pain[6–9]. Nonetheless, the circuit and molecular mechanisms underlying pain-induced adaptations within DA$^{VTA}$ neurons and whether such adaptation is involved in comorbid anhedonia-like behavior (CAB) during chronic pain are poorly understood.

Dopaminergic neurons in the VTA receive inputs from several brain regions. In addition to brain regions known to be engaged in nocifensive information processing (e.g., the parabrachial nucleus (PBN), periaqueductal gray (PAG), and central amygdala (CeA)[10–13]), the dorsal raphe nucleus (DRN), one of the major sources of serotonin (5-HT) throughout the brain, also robustly projects to the VTA[14]. Apart from 5-HT, VTA-projecting DRN neurons also release the co-transmitter glutamate, which mediates reward[15,16]. Although dysfunction of ascending serotonergic neurons has been implicated in depressive disorders, including comorbid depressive symptoms during chronic pain[17], whether maladaptation of glutamatergic transmission of the DRN→VTA reward circuit underlies neuropathic-pain-induced mesolimbic system dysfunction and, by extension, underlies pain and CAB, remains largely unknown.

In the current study, using spared nerve injury (SNI) as a neuropathic pain model, we demonstrate that a glutamatergic circuit from VGluT3$^{DRN}$ to DA$^{VTA}$ neurons (VGluT3$^{DRN}$→DA$^{VTA}$) undergoes maladaptive changes, resulting in reduced DA release in the NAcMed over NAc lateral shell (NAcLat). Activation of the VGluT3$^{DRN}$→DA$^{VTA}$ pathway relieves pain and CAB through D2 receptors and D1 receptors in NAcMed, respectively. Conversely, VGluT3$^{DRN}$→DA$^{VTA}$ inhibition produces hypersensitivity and anhedonia-like behavior in intact mice. Together, our results reveal VGluT3$^{DRN}$→DA$^{VTA}$→NAcMed circuit adaptation in chronic pain and CAB.

## Results

### Neuropathic pain dampens VGluT3$^{DRN}$→DA$^{VTA}$ circuit activity

The mesolimbic dopamine system from the VTA is well known for reward and motivation processing[5,18]. Some studies have suggested that DA$^{VTA}$ neurons also process aversive stimuli, including acute pain[19,20]. To assess whether DA dysfunction occurs following chronic pain, SNI was used as a model of chronic neuropathic pain, wherein long-lasting mechanical hypersensitivity develops after surgery and persists throughout the experimental timeline of 6 weeks (6W) (Supplementary Fig. 1a, b). In agreement with a previous study[17], by 6 weeks post-SNI (post-SNI 6W), mice developed CAB, a key defining feature of depression[21], which was assessed by the sucrose preference test (SPT), whereas CAB was not present during an early stage of neuropathic pain (2W after surgery; post-SNI 2W) (Supplementary Fig. 1c).

To investigate the dynamic activity of DA$^{VTA}$ neurons associated with nerve injury-induced pain-like hypersensitivity and CAB, in vivo fiber photometry synchronized with a video-tracking system was used to monitor Ca$^{2+}$ fluctuations upon hind paw von Frey stimulation and sucrose consumption. First, we injected Cre-dependent adeno-associated virus (AAVs) expressing the fluorescent Ca$^{2+}$ indicator GCaMP6m into the VTA of DAT-Cre mice (Fig. 1a, b). Post hoc staining showed that ~94% of GCaMP6m-expressing neurons were tyrosine hydroxylase (TH) positive, indicative of high specificity of viral targeting for dopaminergic neurons (Fig. 1c). To characterize the activity of DA$^{VTA}$ neurons associated with mechanical hypersensitivity, we delivered innocuous punctate mechanical stimuli (0.4 g von Frey filament), which rarely evoked paw withdrawal responses in pre-SNI mice but caused frequent withdrawal and aversion in post-SNI mice (Supplementary Fig. 2a, b). Interestingly, we observed a significant activation of DA$^{VTA}$ neurons upon von Frey stimulation in pre-SNI mice, whereas the same stimulus evoked an overall inhibition of calcium transients in post-SNI 2W mice (Fig. 1d). We next examined the population-level activity of DA$^{VTA}$ neurons associated with CAB. In pre-

SNI mice, sucrose consumption dramatically increased Ca$^{2+}$ levels, indicating a prominent role of DA$^{VTA}$ neurons in reward and motivation processing (Fig. 1e), as previously reported[22]. The sucrose-induced activation of DA$^{VTA}$ neurons was maintained in post-SNI 2W mice (Supplementary Fig. 3a, b); however, in post-SNI 6W mice with CAB, activation of DA$^{VTA}$ neurons upon sucrose licking was significantly decreased (Fig. 1e). In sham controls, DA$^{VTA}$ neural activities upon von Frey and sucrose stimuli were unaffected at 2 weeks and 6 weeks after surgery, respectively (Supplementary Fig. 3c–e), suggesting that the decrease was not due to signal decays over time. These results establish a potential link between dysfunctional adaptation in DA$^{VTA}$ neural activity, nerve injury-induced pain-like hypersensitivity, and CAB.

Given the observed changes in the population-level activity of VTA$^{DA}$ neurons during neuropathic pain, we next analyzed the underlying physiological mechanisms at the single-neuron level by conducting whole-cell patch-clamp recordings in brain slices (Supplementary Fig. 4a). To visualize dopaminergic neurons, transgenic DAT-tdTOM mice were generated by crossing DAT-Cre mice with Ai14 mice. DAT$^+$ neurons from the VTA displayed a typical hyperpolarization-activated cation current ($I_h$) and a lower excitability profile in response to depolarizing current steps (Supplementary Fig. 4b–d) when compared with DAT$^−$ neurons (Supplementary Fig. 4e), which are likely to be GABAergic[23,24]. Of note, DAT$^+$ neurons in brain slices from post-SNI 2W mice showed a significant decrease in the firing rate compared with sham controls (Supplementary Fig. 4c), which was consistent with previous reports[8]. In contrast, the excitability of DAT$^−$ neurons was comparable between SNI and sham mice (Supplementary Fig. 4e, f). We then examined whether further reduction in the firing rate of DA$^{VTA}$ neurons in post-SNI 6W mice could lead to CAB. Unexpectedly, the DA$^{VTA}$ neurons from post-SNI 6W mice displayed a similar firing rate as those from post-SNI 2W mice (Supplementary Fig. 4c), implying that other factors contribute to the anhedonia-like behavior. Given the unique role of $I_h$ in modulating intrinsic excitability and temporal integration of synaptic input[25], we next analyzed $I_h$, and, intriguingly, observed a drastic reduction of $I_h$ in DAT$^+$ neurons from post-SNI 6W mice compared with sham controls and post-SNI 2W mice (Supplementary Fig. 4d).

Having determined that the population-level activity and intrinsic excitability of DA$^{VTA}$ neurons are blunted during chronic neuropathic pain, we examined whether artificial activation of DA$^{VTA}$ neurons will ameliorate neuropathic pain and CAB. We found that chemogenetic activation of DA$^{VTA}$ neurons was sufficient to rescue mechanical and thermal hypersensitivity of both sexes at post-SNI 2W (Supplementary Fig. 5c, d), which was consistent with a previous study[26]. The pain relief was also observed in post-SNI 6W mice (Supplementary Fig. 5e, f). Moreover, chemogenetic activation of DA$^{VTA}$ neurons robustly alleviated SNI-induced CAB (Supplementary Fig. 5g).

We next explored the potential neural circuit mechanism underlying the neuropathic pain-induced hypoexcitability of DA$^{VTA}$ neurons. First, we conducted a retrograde trans-monosynaptic tracing experiment by injecting Cre-dependent helper viruses into the VTA of DAT-Cre or GAD2-Cre mice (Supplementary Fig. 6a). Two weeks later, EnvA-pseudotyped RV-ΔG-DsRed was injected into the same site. As previously reported[27,28], we detected substantial numbers of DsRed-labeled neurons in the DRN, laterodorsal tegmentum (LDTg), PBN, lateral habenular nucleus (LHb), and CeA of DAT-Cre mice (Supplementary Fig. 6b, d, e). Because glutamatergic projections from the DRN to VTA strongly mediate reward[15,16,29,30], we further evaluated this circuit. Immunofluorescence staining showed that ~62% of DsRed-labeled neurons in the DRN co-localized with VGluT3 (Supplementary Fig. 6b, d), a vesicular transporter that concentrates glutamate into synaptic vesicles[31], which is essential to signal reward[15]. In comparison, fewer than 30% of DsRed-labeled neurons co-expressed VGluT3 in GAD2-Cre mice (Supplementary Fig. 6c, d, f). Anterograde trans-monosynaptic tracing data from C57 mice showed that ~71% of

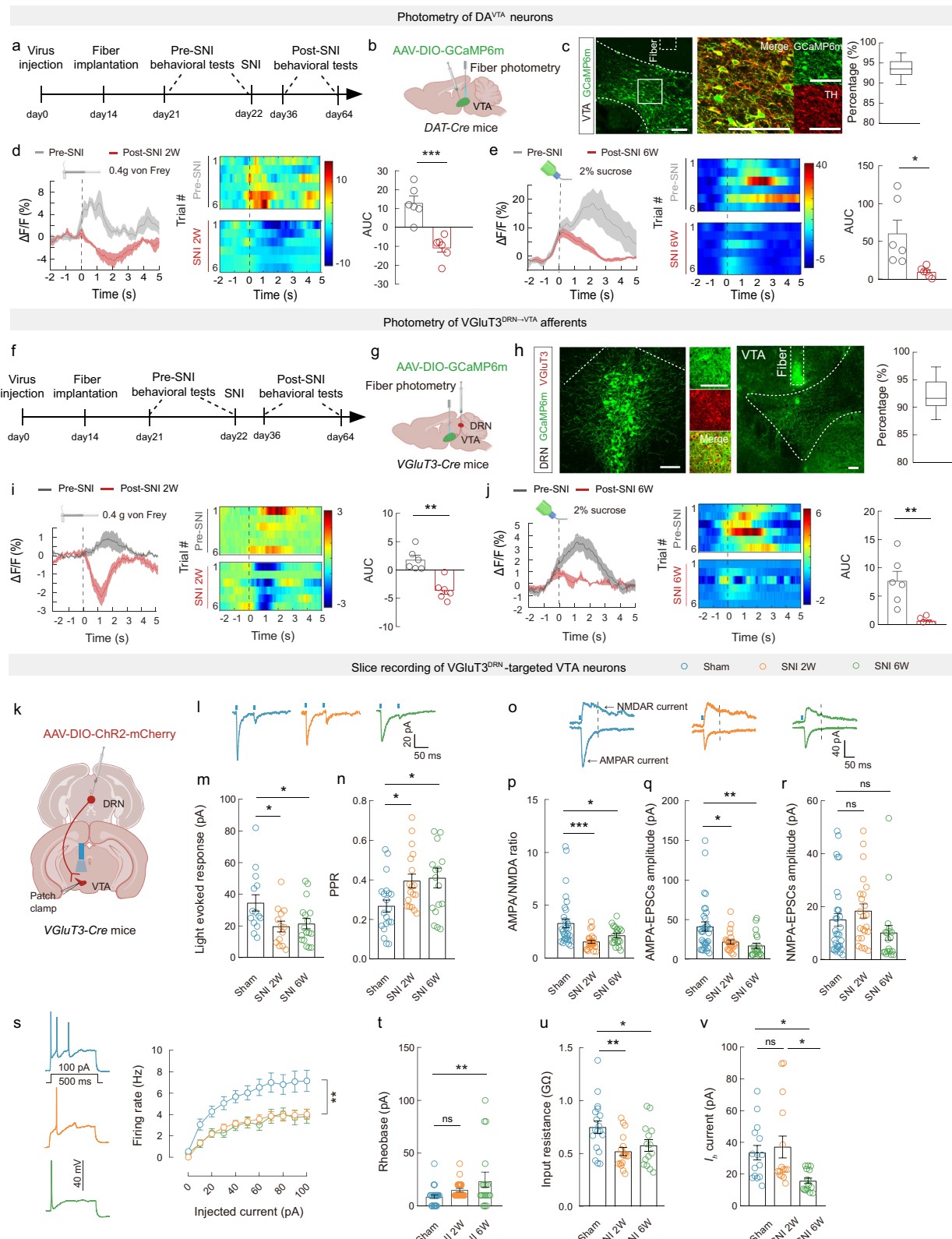

DRN-targeted VTA neurons co-stained with TH antibody (Supplementary Fig. 7a–c), suggesting that the VGluT3$^{DRN}$ neurons send major projections to DA$^{VTA}$ neurons, as illustrated previously[15]. Moreover, we observed prominent innervation of NAc from DRN-targeted VTA neurons (Supplementary Fig. 7d). By injecting Fluoro-Gold (FG) into the VTA of *VGluT3-tdTOM* mice (Supplementary Fig. 8a), we observed that ~60% of FG$^+$ neurons in the DRN contained VGluT3, including ~36%

of FG$^+$ neurons with VGluT3 only and ~23% of FG$^+$ neurons with both 5-HT and VGluT3 (Supplementary Fig. 8b, c). Thus, the majority of VTA-projecting DRN neurons are VGluT3-expressing, some of which co-express 5-HT.

The functional connection between VGluT3$^{DRN}$ and VTA neurons was further characterized by electrophysiology. We injected Cre-dependent AAV-DIO-channelrhodopsin 2 (ChR2)-mCherry into the

**Fig. 1 | Dampened activity of VGluT3$^{DRN}$ → DA$^{VTA}$ circuit in mice with chronic pain. a**, **b**, **f**, **g**, **k** Schematic of the experimental design. **c**, **h** Representative images and percentage of GCaMP6m-expressing neurons that expressed TH (**c**) or VGluT3 (**h**), $n = 9$ sections. Scale bars, 100 µm. **d**, **e**, **i**, **j** Averaged Ca$^{2+}$ responses, heatmaps, and area under the curve (AUC) evoked by von Frey stimulation (**d**: $P = 0.0004$; **i**: $P = 0.096$) and sucrose licking (**e**: $P = 0.0264$; **j**: $P = 0.072$). $n = 6$ mice. **l**−**n** Representative traces (**l**) and statistics of amplitude (**m**: Sham vs SNI 2W $P = 0.0305$; Sham vs SNI 6W $P = 0.0429$) and PPR (**n**: Sham vs SNI 2W $P = 0.0389$; Sham vs SNI 6W $P = 0.0210$). **o**−**r** Representative traces (**o**) and statistics of AMPA/NMDA ratio (**p**: Sham vs SNI 2W $P = 0.0005$; Sham vs SNI 6W $P = 0.0387$), AMPA-EPSCs amplitude (**q**: Sham vs SNI 2W $P = 0.0105$; Sham vs SNI 6W $P = 0.0017$), and NMDA-EPSCs amplitude (**r**: Sham vs SNI 2W $P = 0.7065$; Sham vs SNI 6W $P = 0.3992$).

**s**−**v** Statistics of firing rate (**s**: Sham vs SNI 2W $P = 0.0167$; Sham vs SNI 6W $P = 0.0173$), rheobase (**t**: Sham vs SNI 2W $P = 0.3780$; Sham vs SNI 6W $P = 0.0071$), input resistance (**u**: Sham vs SNI 2W $P = 0.0042$; Sham vs SNI 6W $P = 0.0439$), and $I_h$ (**v**: Sham vs SNI 2W $P > 0.9999$; Sham vs SNI 6W $P = 0.0404$; SNI 2W vs SNI 6W $P = 0.0103$). The groups in panels (**c**, **h**, **l**−**v**) were all from 3 mice. Significance was assessed by two-tailed paired Student's $t$-test in (**d**, **e**, **i**, **j**), one-way ANOVA followed by Bonferroni's multiple comparisons test in (**m**, **n**, **p**−**r**, **t**−**v**), and two-way ANOVA followed by Bonferroni's multiple comparisons test in (**s**). All data are presented as the mean ± s.e.m except for (**c**, **h**) shown as box and whisker plots (medians, quartiles (boxes) and ranges minimum to maximum (whiskers)). *$P < 0.05$, **$P < 0.01$, ***$P < 0.001$, not significant (ns). Detailed statistics are presented in Supplementary Data 1. Created with BioRender.com (**b**, **d**, **e**, **g**, **i**, **j**, **k**).

DRN of *VGluT3-Cre* mice to enable selective activation of VGluT3$^{DRN}$ neural terminals within the VTA (VGluT3$^{DRN→VTA}$ terminals) to examine postsynaptic responses in the VTA (Supplementary Fig. 8d). Current clamp recordings on ChR2-expressing DRN neurons showed that 20 Hz blue light stimulation reliably evoked action potentials without failure, indicating functional viral transduction (Supplementary Fig. 8d). When recordings were performed on VTA neurons, excitation of the VGluT3 afferents by single-pulse light produced fast-onset excitatory postsynaptic currents (fast EPSCs) in 50% of the neurons (22 of 44 neurons) (Supplementary Fig. 8e, i), but rare (1 of 44 neurons, data not shown) if any, fast-onset inhibitory postsynaptic currents (fast IPSCs). The EPSCs were nearly eliminated by bath application of CNQX (10 µM), a selective AMPA/kainate receptor antagonist (Supplementary Fig. 8e). Moreover, EPSCs could be blocked by tetrodotoxin (TTX) and restored by potassium channel blocker 4-aminopyridine (4-AP) (Supplementary Fig. 8f). Low-response jitter of fast EPSCs was also observed (Supplementary Fig. 8g). Thus, VTA neurons receive monosynaptic glutamatergic inputs from VGluT3$^{DRN}$ neurons.

Next, we stimulated the VGluT3$^{DRN→VTA}$ afferents by shedding prolonged blue light (20 Hz for 20 s) and observed that ~32% (14 of 44) of VTA neurons displayed slow outward currents (slow IPSCs) (Supplementary Fig. 8h, i), which were abolished by treatment with ketanserin, a selective antagonist for 5-HT$_{2A}$ and 5-HT$_{2c}$ receptors (Supplementary Fig. 8i). Of note, the 32% of VTA neurons with slow IPSCs concurrently exhibited detectable fast EPSCs, which is consistent with our aforementioned neurochemical study (Supplementary Fig. 8c). Thus, VTA neurons receive both glutamatergic and serotonergic inputs from VGluT3$^{DRN}$ neurons. Furthermore, we specifically recorded DA$^{VTA}$ neurons, which were validated by the presence of $I_h$, and found that ~81% of them displayed fast EPSCs following VGluT3$^{DRN→VTA}$ afferents excitation. In contrast, we only detected fast EPSCs in ~33% of $I_h$-negative neurons (Supplementary Fig. 8j). These data demonstrate that VGluT3$^{DRN}$ neurons make functional connections preferentially with DA$^{VTA}$ neurons.

We next examined whether there is an association between DRN → VTA pathway adaptation with nerve injury-induced pain-like hypersensitivity and CAB. We used fiber photometry to measure calcium dynamics from VGluT3$^{DRN→VTA}$ terminals in response to hind paw mechanical stimulus and sucrose consumption. To this end, *VGluT3-Cre* mice received DRN injection with AAV-DIO-GCaMP6m and implantation of an optical fiber into the VTA (Fig. 1f–h). We observed an elevation in calcium activity upon the delivery of 0.4 g von Frey filament to pre-SNI mice, whereas the same stimulus strongly inhibited calcium activity in post-SNI 2W mice (Fig. 1i). We further found that sucrose licking significantly increased calcium transients of VGluT3$^{DRN→VTA}$ terminals in pre-SNI mice (Fig. 1j), and similar increases were induced by sucrose in post-SNI 2W mice (Supplementary Fig. 9a, b). In contrast, in post-SNI 6W mice that displayed CAB, sucrose licking triggered much lower levels of calcium activity (Fig. 1j). In sham controls, VGluT3$^{DRN→VTA}$ terminal activity was unaffected upon stimulation with von Frey filament or sucrose at 2 weeks and 6 weeks after surgery, respectively

(Supplementary Fig. 9c–e). These findings demonstrate that VGluT3$^{DRN→VTA}$ afferents activity is altered by SNI.

## Neuropathic pain attenuates synaptic strength of the VGluT3$^{DRN}$ → DA$^{VTA}$ circuit

To further explore changes in VGluT3$^{DRN→}$DA$^{VTA}$ circuit activity during chronic neuropathic pain, we performed patch-clamp recordings on VTA neurons (Fig. 1k). Nerve injury did not affect the proportion of synaptic responses evoked by activation of VGluT3$^{DRN→VTA}$ terminals in either post-SNI 2W or 6W mice (Supplementary Fig. 10a, b). Notably, we did not observe any difference in amplitudes of slow IPSCs between pre-SNI mice and post-SNI mice (Supplementary Fig. 10c, d), whereas amplitudes of fast EPSCs recorded from $I_h$-positive DA$^{VTA}$ neurons in SNI mice were significantly attenuated compared with sham controls (Fig. 1l, m).

The compromised glutamatergic transmission may arise from either decreased presynaptic glutamate release from VGluT3$^{DRN→VTA}$ neurons and/or from decreased postsynaptic response to glutamate of VTA neurons. To examine this, we first assessed the paired-pulse ratio (PPR) of light-evoked fast EPSCs, which is known to be inversely correlated with the presynaptic transmitter release. We found that the PPR (ISI 50 ms) of DA$^{VTA}$ neurons was significantly increased in SNI mice compared with sham controls (Fig. 1l, n), implying SNI attenuates presynaptic glutamate release from VGluT3$^{DRN→VTA}$ neurons. Next, we investigated the ratio of AMPA receptor-mediated EPSCs (AMPA-EPSCs) versus NMDA-EPSCs, which is an indicator of postsynaptic plasticity. We detected a smaller AMPA-EPSC:NMDA-EPSC ratio in SNI mice, probably caused by selective attenuation of AMPA-EPSCs (Fig. 1o–r).

We also analyzed the intrinsic excitability of VGluT3$^{DRN}$-targeted postsynaptic DA$^{VTA}$ neurons, which were identified by the presence of $I_h$ and light-evoked fast EPSCs. The input−output plot showed lowered firing rate of DA$^{VTA}$ neurons in brain slices from both post-SNI 2W and 6W mice compared with sham controls (Fig. 1s). Correspondingly, the rheobase, calculated as the minimum current required to evoke an action potential, was increased (Fig. 1t). Consistent with the decreased excitability, a significant decrease in the input resistance was also observed in both post-SNI 2W and 6W mice (Fig. 1u), whereas a significant reduction in $I_h$ was only detected in post-SNI 6W mice with CAB (Fig. 1v). In summary, these data show that SNI dampens synaptic strength of the VGluT3$^{DRN}$ → DA$^{VTA}$ pathway via both presynaptic and postsynaptic mechanisms.

## VGluT3$^{DRN→VTA}$ terminals modulate chronic pain and comorbid anhedonia

Because synaptic transmission of the VGluT3$^{DRN}$ → DA$^{VTA}$ circuit was compromised during chronic neuropathic pain, we postulated that artificial activation of the VGluT3$^{DRN→VTA}$ projection would alleviate pain-like hypersensitivity and CAB. We injected *VGluT3-Cre* mice with AAV-DIO-ChR2-mCherry or AAV-DIO-mCherry into the DRN and implanted optical fibers above the VTA (Fig. 2a, b). We found that optogenetic activation of VGluT3$^{DRN→VTA}$ terminals significantly

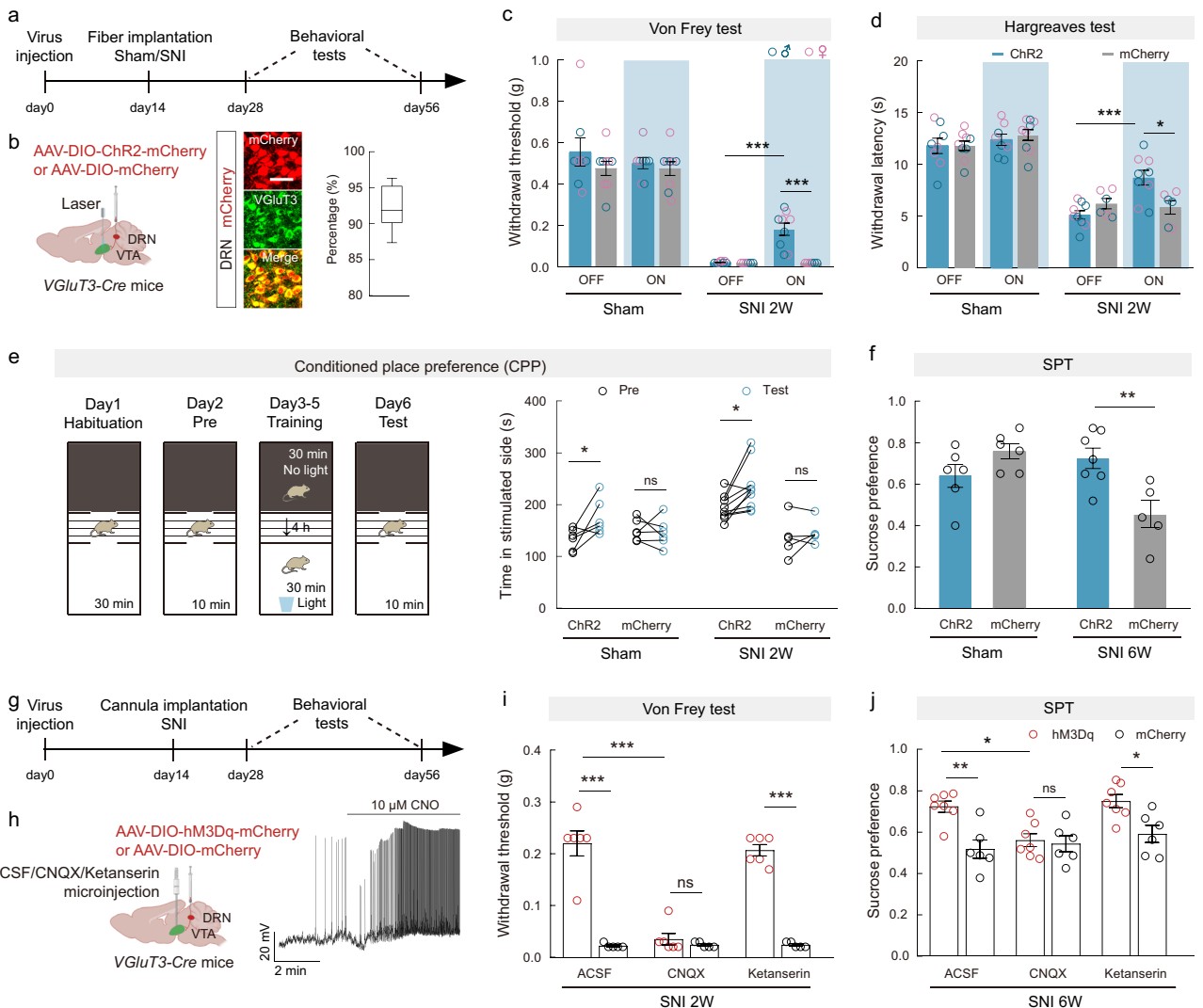

**Fig. 2 | Glutamate is essential for the analgesic and anti-anhedonia effects conferred by VGluT3$^{DRN\rightarrow VTA}$ neural excitation. a, g** Schematic of the experimental design. **b** Schematic diagram of viral injection and optical fiber implantation (left). Representative images (middle) and summary data (right) for the percentage of mCherry-expressing neurons co-localized with VGluT3 immunofluorescence, $n = 9$ sections from 3 mice. Scale bar, 50 μm. **c, d** Mechanical paw withdrawal threshold (**c**: ChR2&ON vs ChR2&OFF $P < 0.0001$; ChR2&ON vs mCherry&ON $P < 0.0001$) and thermal paw withdrawal latency (**d**: ChR2&ON vs ChR2&OFF $P = 0.0005$; ChR2&ON vs mCherry&ON $P = 0.0153$) with (on) or without (off) optogenetic stimulation. Sham&ChR2, $n = 8$; Sham&mCherry, $n = 9$; SNI&ChR2, $n = 9$; SNI&mCherry, $n = 6$. **e** Experimental design of conditional place-preference (CPP) test (left) and quantification of CPP training (right). Sham&ChR2, $n = 6$; Sham&mCherry, $n = 6$; SNI&ChR2, $n = 10$; SNI&mCherry, $n = 5$. Sham&ChR2 $P = 0.0470$; SNI&ChR2 $P = 0.0118$. **f** Preference for sucrose in the SPT. Sham&ChR2,

$n = 6$; Sham&mCherry, $n = 6$; SNI&ChR2, $n = 7$; SNI&mCherry, $n = 5$. $P = 0.0094$. **h** Schematic diagram of viral injection and drug delivery cannula implantation (left), and a representative trace showing depolarization of the hM3Dq-expressing neuron by CNO (right). **i, j** Effects of chemogenetic activation of VGluT3$^{DRN}$ neurons on the von Frey test in post-SNI 2W mice (**i**: $P < 0.0001$) and SPT in post-SNI 6W mice (**j**: ACSF&hM3Dq vs ACSF&mCherry $P = 0.0035$; Ketanserin&hM3Dq vs Ketanserin&mCherry $P = 0.0473$). hM3Dq, $n = 7$; mCherry, $n = 6$. Significance was assessed by two-way ANOVA followed by Bonferroni's multiple comparisons test in (**c, d, f, i, j**) and two-tailed paired Student's $t$-test in (**e**). All data are presented as the mean ± s.e.m except for (**b**) shown as box and whisker plots (medians, quartiles (boxes) and ranges minimum to maximum (whiskers)). *$P < 0.05$, **$P < 0.01$, ***$P < 0.001$, not significant (ns). Details of the statistical analyses are presented in Supplementary Data 1. Created with BioRender.com (**b, h**).

elevated the paw withdrawal threshold in the von Frey test as well as the paw withdrawal latency in the Hargreaves test of post-SNI 2W mice in both sexes (Fig. 2c, d), whereas it had no effect in sham control mice. The pain relief was also observed in post-SNI 6W mice (Supplementary Fig. 11a, b). Locomotion assessed by the open-field test was unaffected by VGluT3$^{DRN\rightarrow VTA}$ terminals activation (Supplementary Fig. 12a, b). These results indicate that excitation of VGluT3$^{DRN\rightarrow VTA}$ terminals can alleviate neuropathic pain, without affecting basal mechanical and thermal nociception.

We next evaluated conditional place preference (CPP), which is correlated with the reward of pain relief, to ascertain whether activating VGluT3$^{DRN\rightarrow VTA}$ terminals in SNI mice induces pain-relief-seeking

behavior. Mice were tested in a three-chamber CPP assay, and blue light stimulation (10-ms pulse at 20 Hz) was continuously delivered for 30 min when mice were restricted in the white chamber, whereas no light was delivered in the black chamber (Fig. 2e). ChR2-injected post-SNI 2W mice and ChR2-injected sham control mice both displayed CPP for the photostimulation-paired chamber, suggesting the involvement of VGluT3$^{DRN\rightarrow VTA}$ projection in mediating reward (Fig. 2e).

We further examined whether CAB in chronic pain can be modulated by VGluT3$^{DRN\rightarrow VTA}$ projection, which releases co-transmitter of glutamate and 5-HT. We found that selective activation of VGluT3$^{DRN\rightarrow VTA}$ terminals in post-SNI 6W mice significantly rescued sucrose preference (Fig. 2f), an indication of CAB relief.

## Glutamate is essential for the analgesic and anti-anhedonia effects conferred by VGluT3$^{DRN→VTA}$ neural excitation

Given the co-transmitter of glutamate and 5-HT in the VGluT3$^{DRN→VTA}$ neurons, we hypothesized that VGluT3$^{DRN}$ neurons could act through glutamatergic and/or serotonergic transmission to the VTA to relieve pain and comorbid anhedonia. To test this, AAV-DIO-hM3D(Gq)-mCherry or AAV-DIO-mCherry was injected into the DRN of *VGluT3-Cre* mice, and a drug delivery cannula was implanted into the VTA (Fig. 2g, h). Then, the ability of VGluT3$^{DRN}$ neuronal activation to relieve neuropathic pain at post-SNI 2 W and comorbid anhedonia at post-SNI 6W was assessed after intra-VTA injection of CNQX or ketanserin, which are selective antagonists of AMPAR and 5-HT$_{2A/2C}$ receptors, respectively. The functional expression of hM3Dq in VGluT3$^{DRN}$ neurons was indicated by neural depolarization by CNO (10 μM) (Fig. 2h). Following i.p. injection of CNO (2 mg/kg), the paw withdrawal threshold in the von Frey test was increased in hM3Dq-expressing post-SNI 2W mice when compared with the control post-SNI 2W mice (Fig. 2i), suggesting that VGluT3$^{DRN}$ neuronal excitation conferred pain relief. Interestingly, the pain relief was selectively abolished by intra-VTA injection of CNQX, but not by ketanserin (Fig. 2i). Sucrose preference assay showed that relief of comorbid anhedonia in post-SNI 6W mice following VGluT3$^{DRN}$ neuronal excitation was also abrogated by CNQX (Fig. 2j). These data suggest that glutamate plays an essential role in both the analgesic and anti-depressive effects exerted by VGluT3$^{DRN→VTA}$ neuronal projection.

## Inactivation of VGluT3$^{DRN→VTA}$ terminals is sufficient to induce hypersensitivity and comorbid anhedonia

The undermined VGluT3$^{DRN}$ → DA$^{VTA}$ transmission during chronic pain prompted us to further examine whether inhibition of VGluT3$^{DRN→VTA}$ terminals could cause pain-like hypersensitivity. We injected *VGluT3-Cre* mice with AAV-DIO-eNpHR-EYFP or AAV-DIO-EYFP into the DRN and implanted optical fibers above the VTA (Fig. 3a, b). Immunostaining and electrophysiological assays confirmed the functional expression of eNpHR in VGluT3$^{DRN}$ neurons (Fig. 3b, c). Mice were then subjected to the von Frey and Hargreaves tests during 594-nm yellow light stimulation. Optical inhibition of VGluT3$^{DRN→VTA}$ terminals robustly induced both mechanical and thermal hypersensitivity in both sexes (Fig. 3d, e). We further assessed whether an aversive affective memory could be evoked using the conventional conditioned place aversion (CPA) behavioral assay, which showed that inactivation of VGluT3$^{DRN→VTA}$ terminals caused CPA (Fig. 3f). Notably, optogenetic inactivation of VGluT3$^{DRN→VTA}$ projection did not affect total distance in the open-field test (Supplementary Fig. 12c, d), indicating that motor performance was unchanged.

Plastic changes in pain processing circuits caused by sustained nociceptive inputs contribute to the pathogenesis of chronic pain[32]. Therefore, we examined whether prolonged inactivation of the VGluT3$^{DRN}$ neurons could induce persistent pain similarly to the SNI model. To this end, we performed repetitive chemogenetic inhibition of the VGluT3$^{DRN}$ neurons by daily injection of CNO (2 mg/kg) for 1 week in *VGluT3-Cre* mice with AAV-DIO-hM4Di-mCherry or AAV-DIO-mCherry injection into the DRN (Fig. 3g, h). The hyperpolarization of hM4Di-expressing neurons in the DRN after CNO treatment confirmed the functionality of the virus (Fig. 3h). We tested CPA 2 days before the first CNO injection. Consistent with our findings in the optogenetic inhibition experiment, chemogenetic inactivation of VGluT3$^{DRN}$ neurons also evoked CPA (Fig. 3i). In addition, 7-days CNO injection induced mechanical hypersensitivity that persisted for more than 6 weeks in the hM4Di group, but not in the mCherry group (Fig. 3j). It is noteworthy that the hM4Di group displayed reduced sucrose preference at 6 weeks after the first CNO injection (post-CNO 6W) (Fig. 3k), which we also observed in post-SNI 6W mice. These data demonstrate that prolonged silencing of VGluT3$^{DRN}$ neurons mimics both chronic pain-like reflexive

hypersensitivity and CAB. Collectively, our data demonstrate that the VGluT3$^{DRN}$→DA$^{VTA}$ circuit modulates chronic neuropathic pain and comorbid anhedonia.

Given that prolonged hypersensitivity was observed at post-CNO 6W, we tested whether synaptic adaption of the VGluT3$^{DRN}$ → DA$^{VTA}$ pathway may occur long after CNO treatment by injecting AAV-DIO-hM4Di-mCherry/AAV-DIO-mCherry and AAV-DIO-ChR2-mCherry simultaneously into the DRN of *VGluT3-Cre* mice (Fig. 3l). Six weeks after CNO injection, we recorded light-evoked EPSC, firing rate, and $I_h$ of DA$^{VTA}$ neurons. Interestingly, amplitudes of light-evoked EPSCs of DA$^{VTA}$ neurons from hM4Di-expressing post-CNO 6W mice showed a significant decrease when compared with the control post-CNO 6W mice (Fig. 3m). In addition, a reduction in firing frequency and $I_h$ were observed in hM4Di group (Fig. 3n, o). These data suggest that the VGluT3$^{DRN}$ → DA$^{VTA}$ circuit undergoes adaptation long after silencing of VGluT3$^{DRN}$ neurons, which is in line with the plastic changes observed in post-SNI 6W mice (Fig. 1m, s, v).

## The VGluT3$^{DRN}$ → DA$^{VTA}$ circuit for neuropathic pain-induced inhibition of DA release outputs to the medial NAc

NAc is the major target for DA$^{VTA}$ neurons processing reward, motivation, and aversive information, including pain. We next investigated whether DRN-projected DA$^{VTA}$ neurons innervate the NAc, and we observed that DRN-targeted VTA neurons sent dense projecting terminals to the NAcMed and NAcLat, whereas sparse mCherry$^+$ fibers were observed in mPFC (Supplementary Fig. 7d). Thus, VTA neurons receiving DRN inputs preferentially project to the NAc but not the mPFC. To anatomically establish the VGluT3$^{DRN}$ → DA$^{VTA}$→NAcMed pathway, we used a cell-type-specific tracing the relationship between input and output (cTRIO) system[33,34] by injecting AAV-retro-DIO-Flp into the medial NAc and injecting AAV-fDIO-TVA-GFP/AAV-fDIO-RVG into the VTA of *DAT-Cre* mice. Two weeks later, RV-EnA-ΔG-DsRed was injected into the VTA. We observed DsRed-labeled neurons in many brain regions, including the DRN (Supplementary Fig. 13b, e). No GFP-positive cells were detected in the DRN, ruling out the possibility of AAV-fDIO-TVA-GFP contaminations during injection (Supplementary Fig. 13b, d). In addition, ~64% of DsRed-labeled DRN neurons expressed VGluT3 (Supplementary Fig. 13c), suggesting that NAcMed-projecting DA$^{VTA}$ neurons indeed receive input from VGluT3$^{DRN}$ neurons.

To examine the functional connectivity of the VGluT3$^{DRN}$ → DA$^{VTA}$→NAcMed and VGluT3$^{DRN}$ → DA$^{VTA}$→NAcLat circuits, we combined optogenetic stimulation of VGluT3$^{DRN→VTA}$ inputs with fiber photometry of the fluorescence dynamics of DA2m, a G-protein-coupled receptor activation-based DA sensor, in the NAcMed and NAcLat. To achieve this, we injected *VGluT3-Cre* mice with AAV-DIO-ChR2-mCherry into the DRN and AAV-hSyn-DA2m into either the NAcMed or NAcLat (Fig. 4a). This allowed us to optostimulate the VGluT3$^{DRN→VTA}$ terminals while simultaneously monitoring DA release in the NAcMed or NAcLat using fiber photometry (Fig. 4b). In pre-SNI animals, there was an instant increase of DA sensor fluorescence both in the NAcMed and the NAcLat after 473-nm laser stimulation (2 s, 20 Hz) (Fig. 4c, d), indicating that activation of the VGluT3$^{DRN→VTA}$ terminals evoked DA release. Intriguingly, in post-SNI animals, DA release upon stimulating VGluT3$^{DRN→VTA}$ terminals was decreased exclusively in the NAcMed (Fig. 4c, d).

To further explore which subregion of the NAc receives the output of the VGluT3$^{DRN}$ → DA$^{VTA}$ circuit for neuropathic pain-induced inhibition of DA release, we measured DA release within the NAcMed or NAcLat in responses to 0.4 g von Frey filament stimulation or sucrose consumption before and after SNI. In pre-SNI mice, von Frey stimulation evoked robust DA release in the

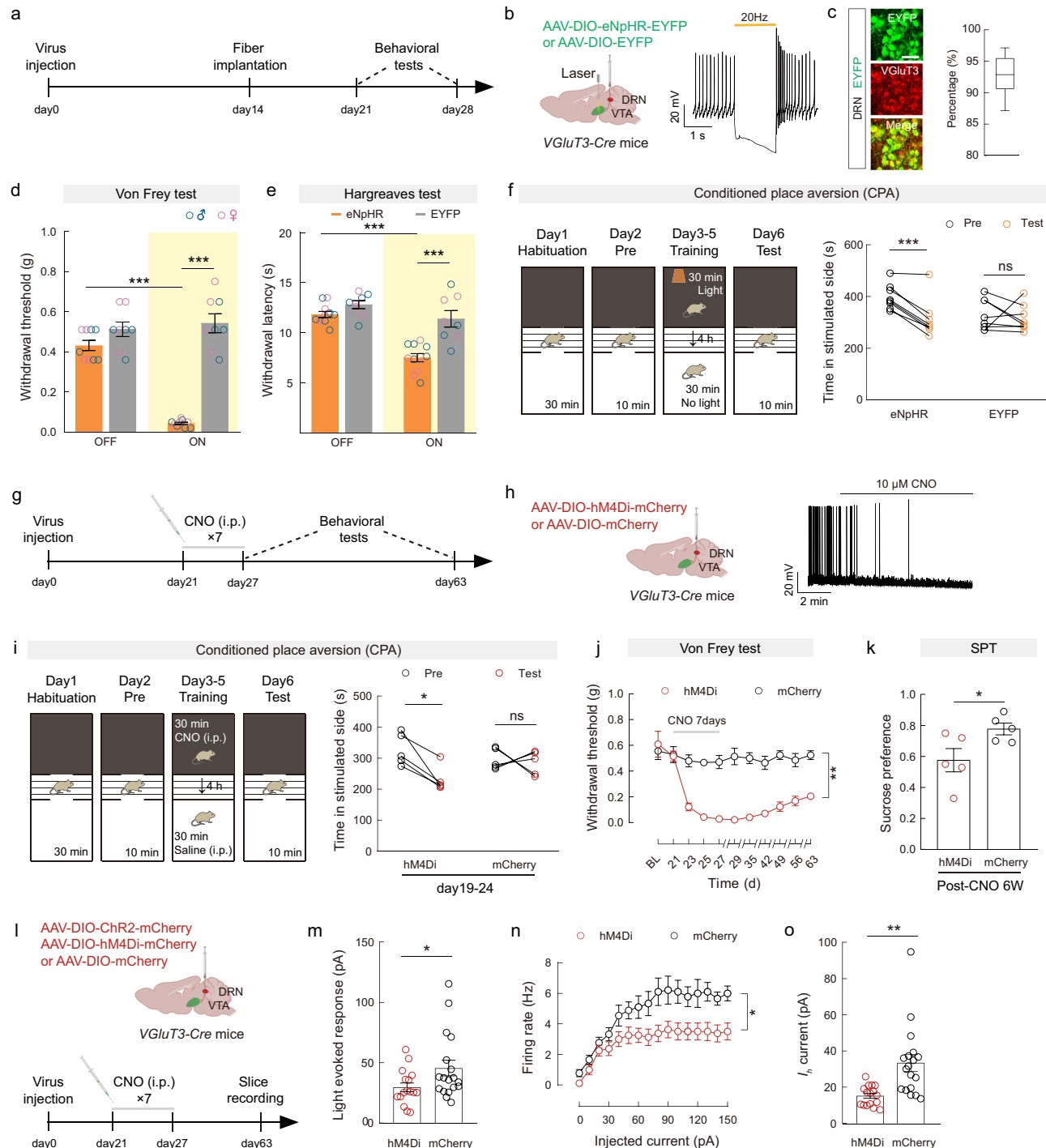

**Fig. 3 | Inactivation of VGluT3$^{DRN→VTA}$ terminals is sufficient to induce reflexive hypersensitivity and comorbid anhedonia. a**, **g**, **l** Schematic of the experimental design. **b** Schematic diagram of viral injection and optical fiber implantation (left), and a representative trace of action potential firing of an eNpHR-expressing neuron during light photostimulation (right). **c** Representative images (left) and summary data (right) for the percentage of EYFP-expressing neurons co-localized with VGluT3 immunofluorescence in the DRN, $n = 9$ sections from 3 mice. Scale bar, 50 μm. **d**, **e** Mechanical paw withdrawal threshold (**d**: eNpHR, $n = 9$; EYFP, $n = 8$. $P < 0.0001$) and thermal paw withdrawal latency (**e**: eNpHR, $n = 11$; EYFP, $n = 8$. $P < 0.0001$) with (on) or without (off) optogenetic stimulation. **f** Experimental design of conditional place aversion (CPA) test (left) and quantification of CPA training (right). eNpHR, $n = 8$; EYFP, $n = 8$. $P = 0.0003$. **h** Schematic diagram of viral injection and optical fiber implantation (left) and a representative trace showing hyperpolarization of the hM4Di-expressing neuron by CNO (right). **i** Experimental

design of CPA test (left) and quantification of training (right). hM4Di, $n = 5$; mCherry, $n = 5$. $P = 0.0112$. **j** Time-course of mechanical paw withdrawal threshold changes. $P = 0.0012$. **k** Preference for sucrose in the SPT, $P = 0.0442$. **m**–**o** Light-evoked EPSCs (**m**: $P = 0.0493$), firing rate (**n**: $P = 0.0332$) and $I_h$ (**o**: $P = 0.0002$) recorded from VGluT3$^{DRN}$-targeted postsynaptic DA$^{VTA}$ neurons. hM4Di, $n = 16$ cells from 3 mice; mCherry, $n = 18$ cells from 3 mice. Significance was assessed by two-way ANOVA followed by Bonferroni's multiple comparisons test in (**d**, **e**, **j**, **n**), two-tailed paired Student's $t$-test in (**f**, **i**), two-tailed unpaired Student's $t$-test in (**k**), and two-tailed Mann–Whitney U test in (**m**, **o**). All data are presented as the mean ± s.e.m except for (**c**), shown as box and whisker plots (medians, quartiles (boxes) and ranges from minimum to maximum (whiskers)). *$P < 0.05$, **$P < 0.01$, ***$P < 0.001$, not significant (ns). Details of the statistical analyses are presented in Supplementary Data 1. Created with BioRender.com (**b**, **h**, **l**).

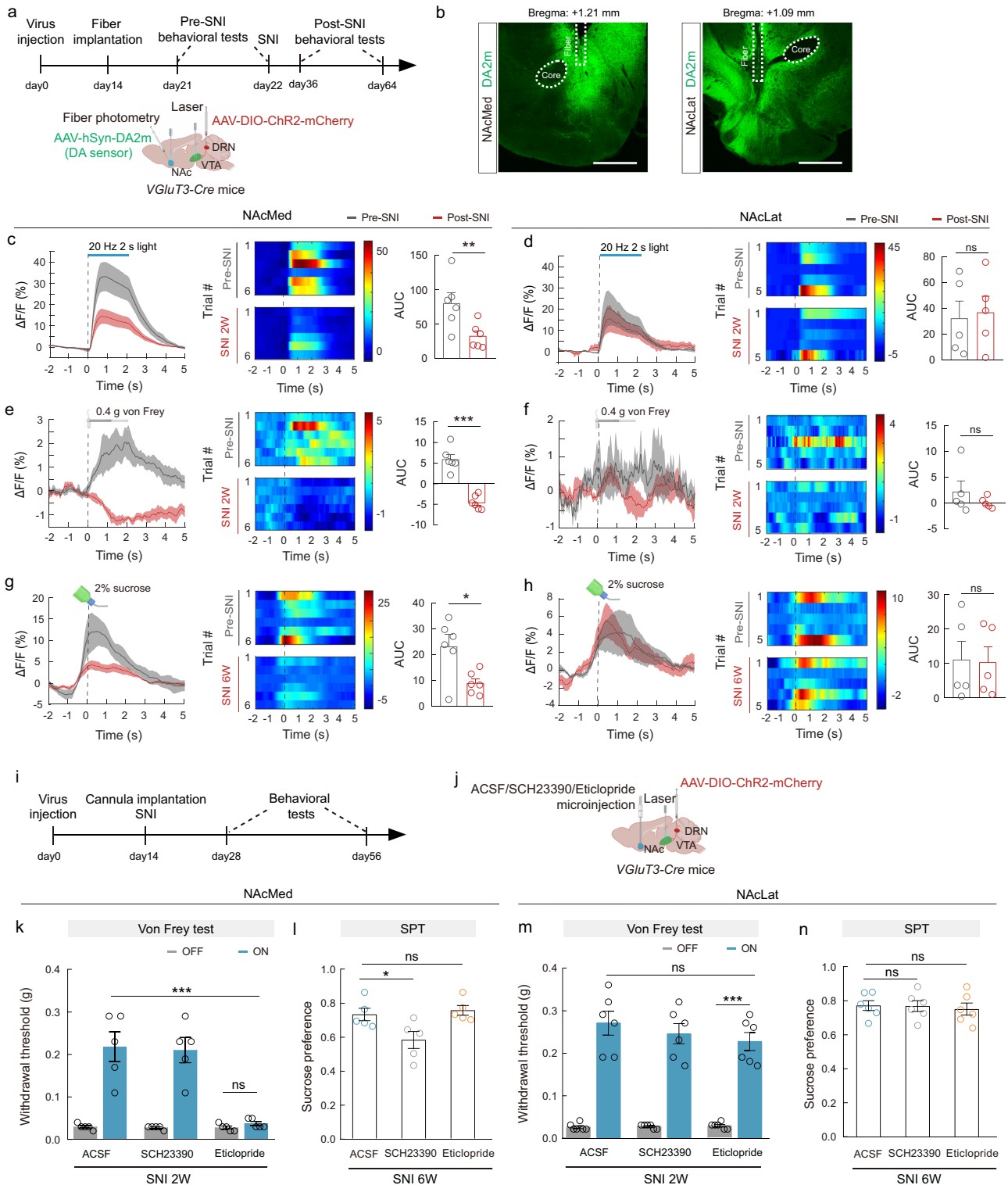

NAcMed. In contrast, subtle DA2m fluctuation was observed in the NAcLat (Fig. 4e, f). Sucrose licking-induced DA release was prominent in both NAcMed and NAcLat (Fig. 4g, h). However, von Frey filament and sucrose consumption-induced DA release were both decreased specifically in the NAcMed of post-SNI 2W and 6W mice, respectively (Fig. 4e, g). In sham controls, DA release upon von Frey and sucrose stimuli were unaffected at 2 weeks and 6 weeks after surgery, respectively (Supplementary Fig. 14). Taken together, we show that medial NAc probably receives the output of the

$VGluT3^{DRN} \rightarrow DA^{VTA}$ circuit for neuropathic pain-induced inhibition of DA release.

## Relief of pain and comorbid anhedonia by $VGluT3^{DRN \rightarrow VTA}$ terminals excitation is via discrete dopaminergic receptors in the NAcMed

D1 and D2 DA receptors (D1R and D2R) are two major types of dopaminergic receptors in the NAcMed. We investigated the requirement for D1R and/or D2R in neuropathic pain and comorbid anhedonia

**Fig. 4 | The VGluT3^{DRN→}DA^{VTA} circuit for neuropathic pain-induced inhibition of DA release outputs to the medial NAc. a, i** Schematic of the experimental design (top) and schematic diagram of viral injection, VTA laser stimulation, and NAc fiber photometry of DA sensor in *VGluT3-Cre* mice (bottom). **b** Representative images showing optical fiber track for fiber photometry recordings in the NAcMed (left) and NAcLat (right). Scale bars, 500 μm. **c–h** Averaged responses (left), heatmaps (middle), and AUC during 0–5 s (right) showing DA2m signals evoked by optogenetic activation of VGluT3^{DRN→VTA} terminals (**c**: $P = 0.0027$, **d**: $P = 0.7926$), 0.4 g von Frey stimulation (**e**: $P = 0.0001$, **f**: $P = 0.3838$) in pre- and post-SNI 2W mice and sucrose licking in pre- and post-SNI 6W mice compared with pre-SNI mice (**g**: $P = 0.0202$, **h**: $P = 0.9069$). NAcMed group, $n = 6$; NAcLat group, $n = 5$. **j** Schematic diagram of DRN viral injection, VTA laser stimulation, and NAc drug delivery

cannula implantation. **k–n** Effects of optogenetic activation of VGluT3^{DRN→VTA} terminals on the von Frey test and SPT with drug infusion into the NAcMed (**k**: ACSF&ON vs Eticlopride&ON $P < 0.0001$; Eticlopride&ON vs Eticlopride&OFF $P > 0.9999$, **l**: ACSF vs SCH23390 $P = 0.037$, ACSF vs Eticlopride $P = 0.8775$. $n = 5$ mice) and NAcLat (**m**: ACSF&ON vs Eticlopride&ON $P > 0.9999$; Eticlopride&ON vs Eticlopride&OFF $P < 0.0001$, **n**: ACSF vs SCH23390 $P = 0.9983$, ACSF vs Eticlopride $P = 0.8684$. $n = 6$ mice). Significance was assessed by two-tailed paired Student's t-test in (**c–h**), two-way ANOVA followed by Bonferroni's multiple comparisons test in (**k, m**), and one-way ANOVA followed by Bonferroni's multiple comparisons test in (**l, n**). All data are presented as the mean ± s.e.m. *$P < 0.05$, **$P < 0.01$, ***$P < 0.001$, not significant (ns). Details of the statistical analyses are presented in Supplementary Data 1. Created with BioRender.com (**a, e–h, j**).

relief. AAV-DIO-ChR2-mCherry or AAV-DIO-mCherry was delivered into the DRN of *VGluT3-Cre* mice. Two weeks later, mice were subjected to cannula implantation into the NAcMed and SNI surgery (Fig. 4i, j). VGluT3^{DRN→VTA} terminals activation-evoked pain and comorbid anhedonia relief were evaluated by intra-NAcMed injection of SCH23390 or eticlopride, which are, respectively, selective antagonists of D1R and D2R. After 473-nm laser stimulation of VGluT3^{DRN→VTA} terminals, the paw withdrawal threshold to von Frey stimulation was elevated in ACSF-infused post-SNI 2W and 6W mice (Fig. 4k; Supplementary Fig. 15a). However, the increased withdrawal threshold was eliminated by Eticlopride, but not by SCH23390 in both post-SNI 2W and 6W mice (Fig. 4k; Supplementary Fig. 15a). In contrast, comorbid anhedonia relief was eliminated by SCH23390, but not by eticlopride in post-SNI 6W mice (Fig. 4l). Of note, relief of neuropathic pain and comorbid anhedonia was not achieved by NAcLat injection of either antagonist (Fig. 4m, n; Supplementary Fig. 15b).

VGluT3^+ DRN neurons also directly project to NAc[29]; therefore, we examined whether the VGluT3^{DRN→NAcMed} pathway contributes to neuropathic pain and CAB relief. We injected *VGluT3-Cre* mice with AAV-DIO-ChR2-mCherry or AAV-DIO-mCherry into the DRN and implanted optical fibers above the NAcMed (Supplementary Fig. 16a, b). Optogenetic activation of VGluT3^{DRN} neural terminals within the NAcMed failed to alleviate both mechanical and thermal hypersensitivity in post-SNI 2W and 6W mice (Supplementary Fig. 16c–f). The CAB was also unaffected (Supplementary Fig. 16g). Thus, we propose that the VGluT3^{DRN} → DA^{VTA}→NAcMed rather than VGluT3^{DRN}→NAcMed pathway plays a predominant role in relieving chronic pain and CAB.

Overall, our study reveals that a glutamatergic circuit from VGluT3^{DRN} to DA^{VTA} neurons undergoes maladaptive changes during neuropathic pain and that the VGluT3^{DRN} → DA^{VTA} circuit modulates pain-like hypersensitivity and CAB via D2R and D1R in NAcMed, respectively.

## Discussion

Management of chronic pain and comorbid depressive symptoms is a significant clinical challenge. In the present study, we identified a crucial role for the VGluT3^{DRN} → DA^{VTA} → D2/D1^{NAcMed} circuit in establishing and modulating chronic neuropathic pain and comorbid anhedonia. We found that chronic pain dampens VGluT3^{DRN} → DA^{VTA} glutamatergic transmission and consequently decreases DA release in a specific subregion of the NAc shell. In addition, VGluT3^{DRN} → DA^{VTA} circuit activation relieved pain-like hypersensitivity and CAB via discrete DA receptors in the NAcMed. Moreover, VGluT3^{DRN} → DA^{VTA} circuit inactivation was sufficient to drive chronic pain-like hypersensitivity and anhedonia-like behavior. These findings provide a neural substrate for treating chronic pain and comorbid anhedonia.

The role of DA^{VTA} neurons in reward processing has long been recognized. However, their roles in encoding aversive information, including pain, remain controversial. Several studies showed that a large proportion of DA^{VTA} neurons are inhibited by acute noxious stimuli[11,35], whereas others demonstrated that a subpopulation of DA^{VTA} neurons is excited by acute pain stimuli[19,20]. The discrepancy could

arise from the anatomical and functional heterogeneity of DA^{VTA} neurons[15,19,20,36]. For example, DA neurons in the ventral VTA are phasically excited by footshocks, whereas those located in the dorsal VTA are inhibited by the same stimulus[19]. In addition, different conditions (physiology vs pathology) could also affect the dopaminergic tone. For instance, both rodents and clinical studies reported that chronic pain induces hypodopaminergic tone[6,8,9,37,38], resulting in anhedonia and depression[2,9,39,40]. However, the circuit mechanisms underpinning chronic neuropathic pain-induced DA neural dysfunction are unclear. In the current work, we demonstrate that dampened activity of the VGluT3^{DRN} → DA^{VTA} pathway may serve as one of the mechanisms underlying DA neural dysfunction in neuropathic pain.

In addition to the reduced firing frequency of DA^{VTA} neurons in neuropathic pain (Supplementary Fig. 4c), which was reported by several groups[8,37,40], we revealed a dynamic stage-dependent downregulation of $I_h$ (Supplementary Fig. 4d) mediated by HCN channels, which are predominantly distributed at the dendrites of DA^{VTA} neurons[41], and which play a pivotal role in temporal integration of synaptic inputs[25]. The impaired $I_h$ would lower the synchrony of neuronal networks from a wide range of synaptic inputs and adversely affect postsynaptic neuronal function[25]. In addition, preclinical studies have shown that enhancing $I_h$ can exert antidepressant effects or achieve homeostatic resilience[42–44]. Thus, our findings suggest potential correlations between the pathophysiological decreases in $I_h$ and the manifestation of stage-dependent CAB during neuropathic pain. Further functional manipulations are required to test such possibilities.

The neural activity of DA^{VTA} neurons is determined by both intrinsic electrophysiological properties as well as sophisticated balancing of excitatory/inhibitory synaptic inputs[45]. Our results provide evidence that reduced excitatory drive from presynaptic VGluT3^{DRN} neurons plays a regulatory role in DA^{VTA} neural activity. The DRN is composed of molecularly and functionally heterogeneous neural subpopulations[34,46], and the role of the DRN serotonergic system in descending pain modulation and relieving depression and several major psychiatric disorders has been well studied[47]. However, how adaptation of the DRN → VTA circuit, which is involved in signaling reward[15,30], may contribute to neuropathic pain and CAB is unknown. Our viral–genetic tracing and electrophysiological data demonstrate that VGluT3^+ neurons in DRN mainly connect with DA^{VTA} neurons (Supplementary Figs. 6 and 7), consistent with a previous report[15]. Our data further show that DRN VGluT3^+ neurons make both glutamatergic and serotonergic synapses with DA^{VTA} neurons (Supplementary Fig. 8), which is consistent with previous findings showing that a subset of DRN neurons project to VTA co-release glutamate and serotonin[15,30,48]. Pathophysiologically, SNI decreased the population-level activity of VGluT3^{DRN→VTA} afferents in response to pain stimuli and sucrose licking as well as weakened VGluT3^{DRN} → DA^{VTA} synaptic connectivity (Fig. 1). Moreover, reduced excitability and a stage-dependent $I_h$ downregulation of DRN-targeted DA^{VTA} neurons were also detected (Fig. 1). Behaviorally, bidirectional manipulation of the VGluT3^{DRN} → DA^{VTA} circuit could alleviate or mimic the SNI-induced pain-like and anhedonia-like behaviors (Figs. 2 and 3).

Pain has both sensory and aversive dimensions. In addition to assessing its sensory component with external stimuli-evoked reflex responses, we used the CPP assay to measure whether the reduction of an aversive state (pain relief) could be achieved following VGluT3$^{DRN}$ → DA$^{VTA}$ circuit excitation and used the CPA assay to assess whether the aversive state could be induced following VGluT3$^{DRN}$ → DA$^{VTA}$ circuit inhibition. Given the role of the VGluT3$^{DRN}$ → DA$^{VTA}$ pathway in mediating reward[15,20], the circuit manipulation itself could cause CPP/CPA. It is thus difficult to conclude that the results of the CPP and CPA experiments reflect changes in pain affection in our study. However, our evidence establishes a compelling correlation between aberrant VGluT3$^{DRN}$ → DA$^{VTA}$ circuit activity and SNI-induced sensory hypersensitivity and CAB. Of note, glutamate from VGluT3$^{DRN→VTA}$ terminals underlies the analgesic and anti-anhedonia effects, but the source from which a subpopulation of VGluT3-expressing neurons releases glutamate remains to be determined. Although 5-HT in the VGluT3$^{DRN}$ → VTA circuit exerted no influence on CAB in our study, the serotonergic adaptation of the DRN→CeA pathway in neuropathic pain has been implicated in comorbid anhedonia[17]. Thus, DRN neurons could regulate comorbid anhedonia in neuropathic pain via different transmitters acting through distinct downstream brain areas. This insight offers hope for strategies to manipulate circuit-based specific receptors for the treatment of chronic pain and CAB. In addition, the retrograde trans-synaptic projections of DRN neurons included the spinal cord, locus coeruleus, periaqueductal gray and basolateral amygdaloid nucleus (Supplementary Fig. 17), which are involved in processing pain information[35,37]. How these DRN-centered circuits cooperate with each other and how these circuits undergo adaptations are important questions to be addressed in future studies to achieve a more detailed understanding of the pathogenesis of chronic pain and CAB.

Dopamine neurotransmission in the nucleus accumbens is commonly considered a primary regulator of motivational drive and is a critical target of drug abuse[18,49,50]. Human studies have demonstrated the promising efficacy of NAc DBS in treatment-refractory pain and major depression[51,52], yet the precise NAc subregions that achieve analgesic and antidepressant effects are ambiguous, especially for comorbid anhedonia in neuropathic pain. In addition, there has been controversy regarding the VGluT3$^{DRN}$ inputs to NAc-projecting DA$^{VTA}$ neurons. A previous study found that activating VGluT3$^{DRN}$ terminals produced larger EPSCs with more frequencies in NAcLat-projecting rather than ventral NAcMed-projecting DA neurons, suggesting that the NAcLat-projecting DA neurons are predominant in promoting reward[20]. In our study, by using fiber photometry, we observed that opto-stimulating VGluT3$^{DRN→VTA}$ terminals caused DA release both in the NAcMed and NAcLat. Interestingly, neuropathic pain evoked region-specific DA alterations preferentially occur in the NAcMed (Fig. 4). Such discrepancy might result from different methodology and targeted populations for recording. We speculate that the VGluT3$^{DRN}$ → DA$^{VTA}$→NAcMed circuit proposed here largely participates in the chronic pain-induced decrease of dopamine release, but not in aversion-related excitation. Moreover, D2R and D1R in the NAcMed mediate the pain and CAB relief by VGluT3$^{DRN→VTA}$ terminals activation, respectively (Fig. 4). These findings advance our understanding of how chronic pain alters the subregion-specific dopaminergic system and related behaviors. Although D1R and D2R in NAc have long been implicated in reward and aversive behaviors, respectively[53], a recent study reported that activation of D2-dopamine receptor-expressing striatal medium spiny neurons (D2-MSNs) in NAc could also drive reinforcement[54]. Therefore, the role of D1-dopamine receptor-expressing MSNs (D1-MSNs) and D2-MSNs in mediating positive and negative motivational valence need to be revisited. The functional divergence between D1-MSNs and D2-MSNs could arise from their different neural circuit connectivity[55,56]. Future studies on

input−output circuits of the two neuronal populations are needed to help delineate their roles in different chronic pain-related behaviors.

A previous study found that activation of the VTA→NAc pathway is sufficient to reverse the pain-induced decrease in motivated behavior without affecting the sensory component of inflammatory pain[9]. We found that the VGluT3$^{DRN}$ → DA$^{VTA}$ circuit modulates reflective hypersensitivity and CAB via DA release in NAcMed. Thus, different manipulations of the mesolimbic system could result in distinct alterations in pain behavior. One plausible explanation could be due to the input−output heterogeneity of midbrain DA neurons[27,57]. In light of this, our data showed that, besides the VGluT3$^{DRN}$, the NAcMed-projecting DA$^{VTA}$ neurons receive inputs from other brain regions, including the LDTg, LC, PAG, LHb (Supplementary Fig. 13d). Since distinct inputs to the NAcMed-projecting DA$^{VTA}$ neurons could affect sensory and motivational behaviors[11,20,22,38,54], it is likely that they impact different components of pain in an input dependent manner. Additionally, varying time points and injury models for measuring pain could also result in different conclusions[58].

Overall, our study demonstrates the dysfunctional adaptation of the glutamatergic VGluT3$^{DRN}$ → DA$^{VTA}$ pathway during chronic pain and CAB development (Supplementary Fig. 18). Our work provides important mechanistic insights regarding the transition from the sensory to the affective components of chronic pain and reveals subregion-specific alteration in DA release and discrete dopaminergic receptors. Therefore, our findings may inform novel therapeutic strategies for managing patients with chronic pain and CAB.

## Methods
### Animals
In all experiments, male and female mice aged 8−10 weeks were used. *C57BL/6J* mice were purchased from Beijing Vital River Laboratory Animal Technology, *VGluT3-Cre* and *Ai14 (RCL-tdTOM)* mice were purchased from Jackson Laboratories, *DAT-Cre*, and *GAD2-Cre* mice were gifts from GQ Bi and Z Zhang, respectively. Mice were kept on a 12-h light/dark cycle (lights on at 7 am) at a stable temperature of $23 \pm 1\,°C$ and a consistent humidity of $50 \pm 5\%$. Food and water were freely available. All animal protocols were approved by the Animal Care and Use Committee of the University of Science and Technology of China.

### Chronic pain model
Chronic neuropathic pain was induced in mice by SNI surgery. Under anesthesia with isoflurane, a skin incision was made on the left thigh, and the muscle was gently separated to explore the sciatic nerve consisting of the sural, common peroneal, and tibial nerves. The common peroneal and tibial nerves were separated from the sural nerves, tightly ligated with nonabsorbent 4-0 chromic gut sutures (Ethicon), and transected together. About 2 mm sections from the nerves were removed. The skin was stitched and disinfected with iodophor. For the sham group, mice were managed in the same manner, but the nerves were not ligated.

### Stereotactic surgery
Under anesthesia induced with an intraperitoneal injection of pentobarbital (20 mg/kg), the mice were fixed in a stereotactic frame (RWD) with a heating pad. Ophthalmic ointment was applied to avoid corneal drying. An incision was made along the midline to expose the skull surface. A volume of 100−300 nl virus was injected in the target region at a rate of 30 nl/min using a pulled glass micropipette connected to a micro syringe pump (KD Scientific). The pipette stayed for an additional 5 min after the injection and was then slowly withdrawn.

For optogenetic manipulation of the DRN-VTA or DRN-NAcMed circuit, we injected rAAV2/9-Ef1α-DIO-hChR2(H134R)-mCherry-WPRE-pA (AAV-DIO-ChR2-mCherry, 4.50E+12 vg/ml, 200 nl), rAAV2/9-Ef1α-DIO-eNpHR3.0-EYFP-WPRE-pA (AAV-DIO-eNpHR-EYFP,

5.36E+12 vg/ml, 200 nl) into DRN (anterior-posterior (AP): −4.50 mm; medial-lateral (ML): 0 mm; dorsal-ventral (DV): −3.15 mm) of *VGluT3-Cre* mice. For chemogenetic behavioral experiments, the rAAV2/9-Ef1α-DIO-hM3D(Gq)-mCherry-WPRE-pA (AAV-DIO-hM3Dq-mCherry, 5.27E+12 vg/ml, 200 nl), the rAAV2/9-Ef1α-DIO-hM4D(Gi)-mCherry-WPRE-pA (AAV-DIO-hM4Di-mCherry, 5.18E+12 vg/ml, 200 nl) viruses were injected into the DRN of *VGluT3-Cre* mice and the VTA (AP, −3.20 mm; ML, −0.45 mm; DV, −4.25 mm) of *DAT-Cre* mice. The rAAV2/9-Ef1α-DIO-mCherry-WPRE-pA (AAV-DIO-mCherry, 5.14E+12 vg/ml, 200 nl), rAAV2/9-DIO-EYFP-WPRE-pA (AAV-DIO-EYFP, 5.14E+12 vg/ml, 200 nl) were used as the controls.

For retrograde monosynaptic tracing, helper viruses that contained rAAV2/9-Ef1α-DIO-RVG-WPRE-pA (AAV-DIO-RVG, 5.29E+12 vg/ml) and rAAV2/9-Ef1α-DIO-EGFP-2a-TVA-WPRE-pA (AAV-DIO-TVA-GFP, 5.56E+12 vg/ml, 1:2, 200 nl) were co-injected into the right VTA of *DAT-Cre* mice or *GAD2-Cre* mice. Two weeks later, the rabies virus RV-ENVA-ΔG-DsRed (2.0E+08TU/ml, 200 nl) was injected into the same site in the VTA. For cTRIO-based retrograde tracing, rAAV2/R-Ef1α-DIO-Flp-WPRE-pA (AAV-retro-DIO-Flp, 5.45E+12 vg/ml, 250 nl) was injected in NAcMed (AP: +1.30 mm, ML: −0.75 mm, DV: −4.50 mm), and a mixture of rAAV2/8-nEf1α-fDIO-EGFP-T2A-TVA-WPRE-pA (AAV-fDIO-TVA-GFP, 5.31E+12 vg/ml) and rAAV2/8-CAG-fDIO-RVG-WPRE-pA (AAV-fDIO-RVG, 5.41E+12 vg/ml, 1:2, 200 nl) was injected into the right VTA of *DAT-Cre* mice. After 2 weeks, 200 nl RV-EnA-ΔG-DsRed was injected into the same site in the VTA. Seven days after the last injection, mice were perfused, and brain slices were prepared for retrograde monosynaptic tracing. For fluorogold retrograde tracing, *VGluT3-tdTOM* mice were injected with retrograde tracer Fluorogold (FG; diluted with saline, 1%, 300 nl) into the VTA. Seven days after injection, brain slices were co-staining with TPH2-specific antibodies to distinguish phenotypes among DRN neurons projecting to the VTA. For the retrograde trans-synaptic tracing experiment, PRV-CAG-EGFP (2.00E+09 PFU/ml, 150 nl) was injected into the DRN of *C57* mice. Four days after the PRV injection, mice were killed to trace the EGFP signal.

For monosynaptic anterograde tracing, rAAV2/1-hSyn-CRE-WPRE-hGH-pA (AAV2/1-Cre, 1.00E+12 vg/mL, 200 nl) was injected into the DRN of *C57* mice to allow the virus to express Cre in the downstream soma. Simultaneously, AAV-DIO-mCherry (200 nl) was injected into the VTA. After 3 weeks of expression, brain slices were prepared for triple tracing or co-staining with TH-specific antibodies or GABA-specific antibodies in the VTA. Unless otherwise stated, all viruses were packaged by BrainVTA.

## In vivo optogenetic manipulations

Two weeks after virus injection, an optical fiber (200 μm, NA = 0.37; Inper) was unilaterally implanted into the right VTA (AP: −3.20 mm, ML: −0.45 mm, DV: −3.80 mm) or NAcMed (AP: +1.30 mm, ML: −0.75 mm, DV: −4.20 mm) of the mice fixed in a stereotactic frame with a heating pad. The delivery of blue or yellow light was controlled using a ThinkerTech stimulator. To assess pain sensitivity following optostimulation, mice were given high-frequency tonic (10 ms duration at 20 Hz) blue (473 nm, 3–5 mW) or yellow (594 nm, 8–10 mW) light stimulation 2 min before von Frey filament or thermal stimuli. For ChR2 activation during the sucrose preference test, tonic blue light stimulation (30 min on/30 min off/30 min on/30 min off) was applied during test day. For CPP/CPA assay, 30 min-tonic blue/yellow light stimulation was applied during training day. An identical stimulus protocol was applied in the control group. By examining the location of the fibers, only mice with the correct locations of optical fibers and viral expression were used for data analysis.

## Fiber photometry

To record calcium fluorescence of VTA DAergic neurons, rAAV2/9-Ef1α-DIO-GCaMP6m-WPRE-pA (AAV-DIO-GCaMP6m, 6.21E+12 vg/ml, 300 nl) was unilaterally injected into the right VTA (AP: −3.20 mm, ML: −0.45 mm, DV: −4.25 mm) of *DAT-Cre* mice. Two weeks after virus injection, an optic fiber (200 μm, 0.37 NA; Inper) was placed into the right VTA (AP: −3.20 mm, ML: −0.45 mm, DV: −3.95 mm). To record the calcium fluorescence of VGluT3$^{DRN}$ terminals in the VTA, 300 nl of AAV-DIO-GCaMP6m was injected into the DRN of *VGluT3-Cre* mice. After allowing for 14 days of virus expression, an optical fiber (200 μm, 0.37 NA; Inper) was implanted into the right VTA. To measure dopamine release, rAAV2/9-hSyn-DA4.4 (AAV-hSyn-DA2m, 5.60E+12 vg/ml, 300 nl) was injected into the NAcMed (AP: +1.30 mm, ML: −0.75 mm, DV: −4.50 mm) and NAcLat (AP: +1.0 mm, ML: −1.80 mm, DV: −4.90 mm) of ChR2-expressing *VGluT3-Cre* mice, and optical fibers were implanted over the NAcMed (AP: +1.30 mm, ML: −0.75 mm, DV: −4.20 mm) and NAcLat (AP: +1.0 mm, ML: −1.80 mm, DV: −4.60 mm) to record change of fluorescence. Photometric recordings were conducted using the fiber photometry recording system (ThinkerTech) 7 days after the fibers-implantation procedures to ensure adequate animal recovery.

Calcium-dependent fluorescence signals were obtained by stimulating neurons and terminals expressing GCaMP6m with laser intensities for the 470 nm wavelength bands (40 μW), and 410 nm signal (20 μW) was further used to correct movement artifacts. Light emission was recorded using an sCMOS Camera, and the values of Ca$^{2+}$ signal changes (ΔF/F) by calculating (F − F0)/F0 (Averaged baseline fluorescence signal recorded) was analyzed by MATLAB. A 5-s window around the stimulation point was analyzed, with the period 2 s before stimulus onset taken as baseline. Ca$^{2+}$ responses during the first six times of the required behavior of each mouse were analyzed.

For fiber photometry recordings in the von Frey test, the filament stimulation (0.4 g) was delivered onto the ipsilateral hind paw (nerve-injured or sham surgery side) for 2 s six times after 30 min habituation, unless there was a withdrawal of the paw, stimulation was terminated at any moment. To avoid sensitization to the stimuli, inter-trial intervals were designed as approximately 120 s. Ca$^{2+}$ responses evoked by von Frey stimulation in SNI/Sham mice compared with pre-SNI/pre-Sham mice were recorded. For the sucrose licking test, mice were habituated with a bottle of 2% sucrose for 24 h followed by water deprivation for 24 h, then were given free access to the bottle of 2% sucrose, Ca$^{2+}$ responses evoked by sucrose licking in SNI/Sham mice compared with pre-SNI/pre-Sham mice were recorded, and fluorescence signals during the first six times of licking behavior per mouse were analyzed. For the DA release test, tonic optostimulation (473 nm, 5 mW, 10 ms duration at 20 Hz) was delivered for 2 s through the fiber implanted in VTA. The fluorescence signals during the first six times were recorded to measure dopamine release in SNI/Sham mice compared with pre-SNI/pre-Sham mice. The onset of each event was tagged by triggering mark key (ThinkerTech), which was time-locked to the fiber photometry system; thus, behaviors and fluorescence signals were captured simultaneously.

## In vivo pharmacological approach

After allowing for 14 days of virus expression, a guide cannula (0.34 mm, RWD) was implanted into areas of interest, which included the VTA (AP: −3.20 mm, ML: −0.45 mm, DV: −3.85 mm), NAcMed (AP: +1.30 mm, ML: −0.75 mm, DV: −4.10 mm) and NAcLat (AP: +1.0 mm, ML: −1.80 mm, DV: −4.50 mm) of the mice fixed in a stereotactic frame. The implant was secured to the skull of the animal with dental cement, and mice were allowed to recover from surgery over 7 days before subsequent behavioral experiments. Microinjections were administered 30 min before testing, and the antagonist dissolved in ACSF (200 nl) was microinjected at a rate of 200 nl/min. To test the participation of glutamatergic or serotonic receptors within VTA in pain and anhedonia relief, we administered microinjections of ACSF,

glutamate AMPA receptor antagonist CNQX (1 μg), or serotonin 5-HT2a and 5-HT2c receptor antagonist ketanserin (1 μg) in the VTA. To evaluate the participation of D1 or D2 receptors within NAc in pain and anhedonia relief, we administered microinjections of ACSF, D1 receptor antagonist SCH23390 (0.1 μg), or D2 receptor antagonist eticlopride (1 μg) in the NAcMed and NAcLat.

## Immunofluorescence and imaging

Mice anesthetized with isoflurane were transcardially perfused with 0.1 M PBS and 4% paraformaldehyde in PBS. Brains were removed and post-fixed in 4% paraformaldehyde at 4 °C overnight, dehydrated in 20%, and then 30% sucrose until they sank. Coronal sections (35 μm) were sliced on a cryostat (Leica CM1950). For immunofluorescence, sections were washed with PBS three times (5 min each) and were incubated with blocking buffer (0.3% Triton X-100, 10% goat serum in PBS) for 1 h at room temperature, then they were incubated (12–24 h at 4 °C) with the primary antibodies anti-TPH2 (1:500, rabbit, Abcam, Cat# ab111828), anti-VGluT3 (1:500, rabbit, Synaptic Systems, Cat# 135203), anti-GABA (1:500, rabbit, Sigma, Cat# A2052), anti-TH (1:1000, rabbit, Emd Millipore, Cat# AB152). After rinsing with PBS, the sections were incubated with the corresponding fluorophore-conjugated secondary antibodies (Goat anti-rabbit Alexa Fluor 488, Goat anti-rabbit Alexa Fluor 594 and Goat anti-rabbit Alexa Fluor 647, 1:500, Jackson ImmunoResearch Labs, Cat# 111-545-003, Cat# 111-585-003 and Cat# 111-605-003) for 2 h at room temperature. Sections were then washed three times with PBS, stained with DAPI, and coverslipped with fluorescent mounting medium. Confocal images were captured on an Olympus FV3000 microscope and analyzed with ImageJ software.

## In vitro electrophysiological recordings

For slices preparation, the mice were deeply anesthetized with isoflurane, and intracardially perfused with ice-cold oxygenated N-Methyl-D-glucamine (NMDG) artificial cerebrospinal fluid (ACSF) that contained (in mM) 93 NMDG, 2.5 KCl, 30 NaHCO3, 1.2 NaH$_2$PO$_4$, 20 HEPES, 25 glucose, 2 thiourea, 3 Na-pyruvate, 0.5 CaCl$_2$, 5 Na-ascorbate, 10 MgSO$_4$ and 3 glutathione (PH 7.3–7.4, 300–305 mOsm). Coronal slices (300 μm) were then removed to ice-cold oxygenated NMDG-ACSF and sectioned on a vibrating microtome (VT1200s, Leica). The brain slices were allowed to recover in oxygenated NMDG-ACSF for 10 min at 32 °C, followed by oxygenated N-2-hydroxyethylpiperazine-N-2-ethanesulfonic acid (HEPES) ACSF that contained (in mM) 92 NaCl, 2.5 KCl, 1.2 NaH$_2$PO$_4$, 3 Na-pyruvate, 30 NaHCO$_3$, 20 HEPES, 25 glucose, 2 thiourea, 5 Na-ascorbate, 2 CaCl$_2$, 2 MgSO$_4$ and 3 GSH (PH 7.3–7.4, 300–305 mOsm) for more than 1 h at 25 °C. Slices were transferred to the recording chamber for electrophysiological recordings. Recordings were performed in oxygenated standard ACSF that contained (in mM) 3 HEPES, 129 NaCl, 1.2 KH$_2$PO$_4$, 3 KCl, 2.4 CaCl$_2$, 1.3 MgSO$_4$, 10 glucose, and 20 NaHCO$_3$ (PH 7.3–7.4, 300–310 mOsm, oxygenated with 95% O$_2$ and 5% CO$_2$) at 32 °C.

Patch-clamp electrophysiology data were analyzed with Clampfit pClamp 10.0 software using MultiClamp 700B (Molecular Devices) and Digidata 1550B (Molecular Devices), digitized at 5 kHz, and filtered at 2 kHz. The pipette (6–8 MΩ) was pulled by a micropipette puller (P-1000, Sutter instrument) and filled with the internal solution that contained (in mM): 130 K-gluconate, 5 KCl, 4 Na$_2$ATP, 0.5 NaGTP, 20 HEPES, 0.5 EGTA, (PH 7.28, 290–300 mOsm). To measure $I_h$ currents, neurons were voltage-clamped at −70 mV and stepped to −120 mV in increments of 10 mV. To record an input–output curve of neuronal excitability, 500 ms pulses with 10 pA command current steps were injected from −60 to +150 pA, and the numbers of spikes were quantified for each step. The rheobase was defined as the minimum current required to evoke an action potential.

To examine the monosynaptic nature of eEPSCs and eIPSCs in VTA neurons following light activation of VGluT3$^{DRN}$ inputs, neurons were held at −70 mV and 0 mV, respectively. TTX (1 μM) was used to block action potential-based synaptic transmission. Both TTX (1 μM) and 4-AP (100 μM) were used to restore monosynaptic current. The response jitter was calculated by measuring the standard deviation of the latency values of consecutive EPSCs for VTA neurons. For evaluating synaptic identities, AMPA-mediated fast EPSCs were blocked by the bath application of CNQX (10 μM), and repetitive light stimulation (20 s, 20 Hz)-evoked slow IPSP was largely abolished by the addition of the 5-HT receptor antagonist ketanserin (10 μM). For evaluating the presynaptic mechanism, paired pulses (10 ms duration) with an interval of 50 ms (ISI 50 ms) were delivered, and the PPR was calculated as the amplitude ratio EPSC$_2$/EPSC$_1$.

To measure the AMPA/NMDA current ratio, the pipettes were filled with the intracellular solution that contained (in mM) 135 Cs methanesulfonate, 10 KCl, 1 MgCl$_2$, 2 QX-314, 0.3 GTP-Na, 4 ATP-Mg, 0.2 EGTA, and 20 phosphocreatine, (PH 7.28, 290–300 mOsm). Cells were first clamped at −70 mV, and the AMPA receptor-mediated EPSCs were recorded. To record the NMDA receptor-mediated EPSCs, cells were clamped at +40 mV in the presence of bicuculline, and EPSCs were determined 50 ms after the peak of the current response.

## Sucrose preference test (SPT)

Mice were habituated with two identical bottles of 1% sucrose for 48 h followed by 48 h of water individually, then subjected to water deprivation for 24 h. For the test day, the mice were given access to a two-bottle choice for 2 h, one containing water and the other containing 1% sucrose, and the bottle positions were switched after 1 h. Sucrose preference was calculated as the proportion of 1% sucrose in the total liquid consumed. For chemogenetic experiments, a single injection of CNO (2 mg per kg) was given 30 min before the behavior tests.

## Open-field test (OFT)

The open-field arena was a square box (40 × 40 cm) within a sound-attenuated room. Mice were allowed to freely explore their surroundings, and the total distance traveled was measured by Smart3 software. To assess the effect of optogenetic activation or inhibition on locomotor activity, mice were tested for a 5-min session.

## Von Frey test

The von Frey filaments test was used to assess the onset and maintenance of mechanical allodynia. Mice were individually placed in a transparent plastic box (6.5 × 6.5 × 6 cm) to habituate for about 30 min for 3 continuous days before the behavior tests. The von Frey filaments (bending force ranging from 0.02 to 2 g, North Coast Medical), which increased in stimulation value, quantified mechanical allodynia by measuring the hind paw response. A positive response included brisk paw withdrawal, flinching, licking, or shaking. Using Dixon's up-down method[59], the stimulus producing 50% likelihood of a positive response was determined and taken as the paw withdrawal threshold. For the mechanical withdrawal frequency test in Supplementary Fig. 2a, filaments were applied to the lateral region of the hind paws. Withdrawal frequencies were recorded (5 applications per filament, each applied 30 s apart). For chemogenetic experiments, a single injection of CNO (2 mg per kg) was given 30 min before the behavior tests.

## Hargreaves test

Thermal withdrawal latency was measured with Hargreave's Apparatus (Model390; IITC Life Science Inc, Woodland Hills, CA). Mice were

placed individually in chambers on top of a glass platform and allowed to habituate for at least 60 min before testing. A radiant heat beam was focused onto the ipsilateral hind paw (nerve-injured side), and paw withdrawal latency was recorded with five trials per animal. Five trials were conducted with a minimal interval of 5 min. The maximum and minimum PWL were excluded to minimize the variation, and an average of the remaining three trials was calculated for each mouse. A cut-off latency of 20 s was set to avoid tissue damage. The experimental timeline was the same as the von Frey test.

### RTPA test
Two weeks after SNI surgery, the mice were placed in a custom-made chamber with two distinct visual compartments. The mice moved freely between two chambers for 10 min (Pre). Then we assigned one side as the filament (0.4 g) stimulation side for the following 10-min test. Initially, the mouse was placed in the non-stimulated side of the chamber, and every time the mouse crossed to the stimulation side, the filament stimulation was delivered until the mouse crossed back into the non-stimulation side. In the post-stimulation phase, the mouse freely explored two chambers again. The locations of mice were tracked by a video camera positioned above the chamber.

### CPP test
The conditioned place-preference test consisted of 6 days and was performed in the box consisting of two unique conditioning chambers with a neutral middle chamber. On day 1 (Habituation), the mice were habituated to the box for 30 min. On day 2 (Pre), individual mice were placed in the neutral chamber and allowed to freely explore the entire box for 10 min, then showed a small preference for one of the two chambers. On days 3–5 (Training), mice were confined to preferred chambers for 30 min without light. Four hours after the conditioning (No light), mice were individually conditioned in the non-preferred chamber for 30 min with blue light stimulation (473 nm, 3–5 mW, 10 ms duration at 20 Hz). On day 6 (Test), like day 2, mice were placed in the middle chamber and allowed to explore the entire box for 10 min. The time spent in a non-preferred chamber was analyzed to determine whether optogenetic activation was able to produce CPP.

### CPA test
To test the roles of optogenetic inhibition of VGluT3$^{DRN \rightarrow VTA}$ terminals in aversive aspects, the same procedures were followed as the CPP assay, except that the preferred chamber was assigned as yellow light (594 nm, 8–10 mW, 10 ms duration at 20 Hz)-paired side and the time spent in the preferred chamber was analyzed. Similarly, during the CPA assay for chemogenetic inhibition of VGluT3$^{DRN}$ neurons, the preferred chamber was re-designated as a CNO-paired chamber, in which the animal was injected with CNO (2 mg per kg).

### Quantification and statistical analysis
Data were analyzed with GraphPad Prism v.8.0.1, Olympus FV10-ASW 4.0a Viewer, ImageJ, Clampfit software v.10.0, and MATLAB software. All experiments and data analyses were conducted double-blind, including fiber photometry, immunofluorescence, electro-physiology, and behavioral analyses. For representative micro-graphs, each experiment has been repeated at least three mice with multiple brain slices in the manuscript with consistent results. No data were excluded from the analyses. Experimental and control animals were randomized throughout the study. Student's $t$-tests (paired and unpaired), one-way or two-way ANOVA tests followed by Bonferroni's test for multiple comparisons and Mann–Whitney U test were used to determine statistical differences. All data are presented as the mean ± s.e.m. Statistical results in the figures are presented as symbols $*P < 0.05$, $**P < 0.01$, $***P < 0.001$. Source data are provided as a Source Data file.

### Reporting summary
Further information on research design is available in the Nature Portfolio Reporting Summary linked to this article.

## Data availability
Source data are provided with this paper. There are no restrictions on data availability in the manuscript. Source data are provided with this paper.

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

## Acknowledgements

We thank Dr. Qiufu Ma at Westlake University for critical comments. This study was supported by the National Natural Science Foundation of China (grants 32271048 to Y.Z., 32070999 to Y.Z., and U20A20357 to X.F.L.), Anhui Provincial Natural Science Foundation (grant 2008085J16 to Y.Z.), and the Fundamental Research Funds for the Central Universities (WK2070210004 to Z.Z. and WK9110000056 to X. F. L.).

## Author contributions

X.Y.W. and W.B.J. designed the studies and conducted most of the experiments and data analysis. X.Y.W. wrote the first draft. X.X., R.C., L.B.W. and X.J.S generated molecular and behavioral data. P.F.X. and X.Q.L. managed the mouse colonies used in this study. J.W. and Y.Y.L.

were involved in the overall design of the study. X.Y.S. was involved in the revision of the manuscript. Z.Z., X.F.L., and Y.Z. were involved in the overall design of the project, data analysis, and editing of the final manuscript.

## Competing interests

The authors declare no competing interests.
