## [Peer Review File · Nature Communications]

A glutamatergic DRN–VTA pathway modulates neuropathic pain and comorbid anhedonia-like behavior in miceREVIEWER COMMENTS

Reviewer #1 (Remarks to the Author):

The manuscript by Wang et al. provides novel evidence for downregulation of a monosynaptic VGLUT3DRN \rightarrow DAVTA circuit involved in SNI-induced pain and CDB (measured with sucrose preference test). The paper incorporates an elegant combination of viral tracing, fiber photometry, chemogenetics, optogenetics, and electrophysiology to demonstrate necessity and sufficiency of VGLUT3DRN \rightarrow DAVTA activity in suppressing hypersensitivity and CDB provoked by SNI, and characterize a novel mechanism by which VTA DA activity is suppressed in conditions of pain. The manuscript further expands on these findings by implicating downstream effects localized to the NAcMed and regulated by D2R at early timepoints related to hypersensitivity and D1R at later timepoints when CDB is observed; although, it is unclear how either manipulation impacts hypersensitivity at 6 weeks. Overall, the experiments are adequately powered and include the appropriate controls. The claims are reasonably concluded and not overstated in the context of the results.

The authors are commended on their thorough investigation of SNI and treatment manipulations (CNO) at both 2 and 6 weeks, although it is unclear why some manipulation effects on pain sensitivity were not also assessed at 6 weeks, which could warrant further justification. Similarly, experiments using a within subjects design to assess the effects of SNI, particularly on evoked calcium transient activity, should include sham control groups or otherwise a measure that demonstrates the stability of emitted fluorescence at baseline or an alternative task where SNI-induced effects have been demonstrated through other experiments to be unaffected. Otherwise the potential loss of signal over time cannot be ruled out. In addition methodological details outlining how laser stimulations in the RTPT and photometry signals were time locked to Von Frey filament stimulations and bouts of licking would aid in the interpretation of the findings. Finally, please also indicate whether sex was accounted for in the experiments – though there is mention that experiments were carried out in both sexes it should be clarified if efforts were made to include an equal representation within experiments. Collectively, the studies capture a novel mechanism involved in chronic pain which are expected to advance the field of pain research.

Reviewer #2 (Remarks to the Author):

Uncovering the neuronal circuitry mediating pain may provide opportunities to develop new therapies for pain treatment and limit the development of comorbid disorders. Here, the authors used multiple techniques, including fiber photometry, electrophysiology, optogenetics, chemogenetics, and rabies virus tracing, to investigate the involvement of Vglut3DRN-DAVTA -NacMed circuit in the regulation of chronic pain and comorbid depression. However, I have major concerns about several places.

Major concerns:

1. Several conclusions and figures are very similar to previous studies. For example,
a) The conclusion “VTA neurons from SNI had lower firing rates” in Fig. 1 has been reported in Ren et al., 2016.

b) The conclusion “most DR neurons projecting to the VTA express VGluT3” in Fig. 2 has been reported in Qi et al., 2014. Rabies tracing of dopamine neurons in the VTA has been reported by multiple groups as well.

Repeated work in comparison with previous studies also presented in several supplementary figures. For example, findings on “inflammatory pain decreased the activity of ventral tegmental area (VTA) dopamine (DA) neurons” and “the decreased activity of DA neurons was associated with reduced motivation for natural rewards, consistent with anhedonia-like behavior” has been reported in details in Markovic et al., 2021.

I strongly suggest the authors compress the paper into four figures, and only focus on the actual novel findings. And please cite and discuss relevant work thoroughly. Two-bottle test is a standard assay for anhedonia-like behavior, without other behavioral assays supporting the conclusion on comorbid depressive-like behavior (such as learned helplessness, forced swimming, and etc.), please just say anhedonia-like behavior. Using a different phrasing is not going to change the fact about the novelty of the work, but only reduces accuracy.

2. The authors stated that “our findings reasoned a causal connection between the pathophysiological decreases in Ih with stage-dependent CBD manifestation during neuropathic pain” (line 429). However the data presented in this study is only supporting a correlation between the two subjects but not causal relationships.

A causal relationship between “pathophysiological decreases in Ih with stage-dependent CBD manifestation during neuropathic pain” can only be established if gain-of-function and loss-of-function studies have been done with Ih and the consequences are analysed.

3. As mentioned in the manuscript that Vglut3DRN-DAVTA pathway promotes reward, it’s not surprising that inhibiting this pathway causes CPA. Thus, it’s not good evidence to link this pathway with ongoing pain.

CPA is not “an effective assay for measuring ongoing pain” (line 321). It never is. Thus conclusion of “prolonged silencing of Vglut3DRN neurons mimic both chronic pain-like hypersensitivity and CDB” (line 342) is unfounded.

4. If the authors want to establish the pathway of Vglut3DRN->DAVTA->Medial Nac, they need to use cTRIO (Schwarz et al. 2015), to show that DAVTA neurons projecting to Medial Nac indeed receive input from Vglut3+ DRN neurons.

And because Vglut3+ DRN neurons also directly project to Nac, the authors need to show the Vglut3DRN->DAVTA->Medial Nac pathway is the predominant one for the findings but not Vglut3DRN->Nac.

5. There are also some inconsistencies between parts of the results in this manuscript and previous publications. Please provide more information.

a. Whether VTA DA neurons are activated by aversive stimuli. Ungless et al., 2004 reported that VTA dopamine neurons are specifically excited by reward, while a population of nondopamine neurons in the

VTA is excited by aversive stimuli. Markovic et al., 2021 also reported that pain suppresses VTA DA neuron activity. de Jong et al., 2019 suggested a subpopulation, vNAcMed-projecting VTA DA neurons are activated by the initial foot shock. However, that experiment was conducted by terminal recording rather than soma recording.

b. Whether the medial Nac is the target of the Vglut3DRN-DAVTA pathway. de Jong et al., 2019 reported a different statement that DR VGLUT3 Inputs to VTA activate NAcLat-Projecting DA neurons.

c. Whether VTA-Nac pathway regulates hyperalgesia. In the study of Markovic et al., 2021, they reported that activation of VTA–Nac-projecting DA neurons did not alter hyperalgesia induced by complete Freund’s adjuvant (CFA) injection.

Mirror concerns:

1. Would it be possible that the decreased calcium responses (especially the response to sucrose drinking in SNI 6W) in Fig. 1 and 3 are caused by signal decay?
2. Please add asterisk (between SNI 2W and SNI 6W) in Fig. 1j to support the statement in Page 5 of the manuscript that “dramatic Ih reduction in DAT neurons from SNI 6W mice ...”
3. Please provide more detailed method of optostimulation in Fig. 4c (phasic or tonic? Coupled with filament stimulation or not?)
4. Fig. 5i, prolonged hypersensitivity was observed 6W after CNO injection. Curious whether any neural adaption (e.g., dampened activity of Vglut3DRN-DAVTA pathway) also can be seen long after the termination of CNO injection.
5. What is the reference of line 48-51?
6. Please change the title of Figure 2 with a more specific one.
7. What are the references of line 442-443, “the functional glutamatergic and serotonergic synapse ... by electrophysiology”?
8. Please find a native speaker to help with the writing.

Reference:

- de Jong, J. W., Afjei, S. A., Pollak Dorocic, I., Peck, J. R., Liu, C., Kim, C. K., Tian, L., Deisseroth, K., & Lammel, S. (2019). A Neural Circuit Mechanism for Encoding Aversive Stimuli in the Mesolimbic Dopamine System. *Neuron*, 101(1), 133-151.e7. <https://doi.org/10.1016/j.neuron.2018.11.005>
- Markovic, T., Pedersen, C. E., Massaly, N., Vachez, Y. M., Ruyle, B., Murphy, C. A., Abiraman, K., Shin, J. H., Garcia, J. J., Yoon, H. J., Alvarez, V. A., Bruchas, M. R., Creed, M. C., & Morón, J. A. (2021). Pain induces adaptations in ventral tegmental area dopamine neurons to drive anhedonia-like behavior. *Nature Neuroscience*, 24(11), 1601–1613. <https://doi.org/10.1038/s41593-021-00924-3>
- Qi, J., Zhang, S., Wang, H. L., Wang, H., de Jesus Aceves Buendia, J., Hoffman, A. F., Lupica, C. R., Seal, R. P., & Morales, M. (2014). A glutamatergic reward input from the dorsal raphe to ventral tegmental area dopamine neurons. *Nature Communications*, 5. <https://doi.org/10.1038/ncomms6390>
- Ren, W., Centeno, M. V., Berger, S., Wu, Y., Na, X., Liu, X., Kondapalli, J., Apkarian, A. V., Martina, M., & Surmeier, D. J. (2016). The indirect pathway of the nucleus accumbens shell amplifies neuropathic pain. *Nature Neuroscience*, 19(2), 220–222. <https://doi.org/10.1038/nn.4199>
- Schwarz LA, Miyamichi K, Gao XJ, Beier KT, Weissbourd B, DeLoach KE, Ren J, Ibanes S, Malenka RC, Kremer EJ, Luo L. Viral-genetic tracing of the input–output organization of a central noradrenergic

circuit. *Nature*. 2015 Aug 6;524(7563):88-92.

Ungless, M. A., Magill, P. J., & Bolam, J. P. (2004). Uniform Inhibition of Dopamine Neurons in the Ventral Tegmental Area by Aversive Stimuli. *Science*, 303(5666), 2040–2042.

<https://doi.org/10.1126/science.1093360>

REVIEWER COMMENTS

Reviewer #1 (Remarks to the Author):

We took the liberty to subdivide this reviewer's comments into 5 points:

1. The manuscript by Wang et al. provides novel evidence for downregulation of a monosynaptic $VGLUT3^{DRN} \rightarrow DA^{VTA}$ circuit involved in SNI-induced pain and CDB (measured with sucrose preference test). The paper incorporates an elegant combination of viral tracing, fiber photometry, chemogenetics, optogenetics, and electrophysiology to demonstrate necessity and sufficiency of $VGLUT3^{DRN} \rightarrow DA^{VTA}$ activity in suppressing hypersensitivity and CDB provoked by SNI, and characterize a novel mechanism by which VTA DA activity is suppressed in conditions of pain. The manuscript further expands on these findings by implicating downstream effects localized to the NAcMed and regulated by D2R at early time-points related to hypersensitivity and D1R at later timepoints when CDB is observed; *although, it is unclear how either manipulation impacts hypersensitivity at 6 weeks*. Overall, the experiments are adequately powered and include the appropriate controls. The claims are reasonably concluded and not overstated in the context of the results.

RESPONSE: First, we sincerely thank this referee for his/her positive evaluation of our work and constructive suggestions towards improving our study. To address whether and how D1R and D2R within NAcMed impact pain hypersensitivity at late time-points, we measured mechanical allodynia at 6 weeks after SNI upon simultaneous $VGLUT3^{DRN} \rightarrow DA^{VTA}$ activation and $D1R/D2R^{NAcMed}$ blockage. Our data demonstrated that D2R within the NAcMed plays an indispensable role in alleviating hypersensitivity at 6 weeks after SNI (Response Fig. 1, and also see new Supplementary Fig. 15).

We have incorporated these data into the revised manuscript.

Response Fig 1. D2 receptors within NAcMed contribute to pain relief through VGlut3^{DRN}→DA^{VTA} circuit in post-SNI 6W mice. a, b, Effects of optogenetic activation of VGlut3^{DRN}→VTA terminals on punctate mechanical hypersensitivity with drug infusion into the NAcMed (a) or NAcLat (b) in post-SNI 6W mice. Significance was assessed by two-way ANOVA followed by Bonferroni's multiple comparisons test. All data are presented as the mean ± s.e.m. **P* < 0.001, not significant (ns). Details of the statistical analyses are presented in Supplementary Table 1. See also new Supplementary Fig. 15.**

2. The authors are commended on their thorough investigation of SNI and treatment manipulations (CNO) at both 2 and 6 weeks, although it is unclear why some manipulation effects on pain sensitivity were not also assessed at 6 weeks, which could warrant further justification.

RESPONSE: We appreciate this suggestion to measure pain sensitivity at 6 weeks. To address this concern, we have performed new experiments to assess the effect of VTA^{DA} neural activation using chemogenetics (Response Fig. 2 a, b, and also see new Supplementary Fig. 5e, f), VGlut3^{DRN}→DA^{VTA} circuit activation using optogenetics (Response Fig. 2 c, d, and also see new Supplementary Fig. 11a, b), and impact of D1/D2^{NAcMed/NAcLat} manipulation (Response Fig. 1, and also see new Supplementary Fig. 15) on the hypersensitivity at 6 weeks after SNI. We found, in all cases, manipulation effects on pain sensitivity were consistent between 2 and 6 weeks.

We have incorporated these data to the revised manuscript.

Response Fig 2. Effects of distinctive manipulations on chronic pain hypersensitivity at 6W post-SNI. a, b, Mechanical paw withdrawal threshold (a), thermal paw withdrawal latency (b) of hM3Dq-mCherry-expressing and mCherry-expressing post-SNI 6W mice with saline or CNO treatment. **c, d,** Mechanical paw withdrawal threshold (c), thermal paw withdrawal latency (d) of ChR2-mCherry-expressing and mCherry-expressing Sham or post-SNI 6W mice with (on) or without (off) optogenetic stimulation. Significance was assessed by two-way ANOVA followed by Bonferroni's multiple comparisons test. All data are presented as the mean \pm s.e.m. * $P < 0.05$, *** $P < 0.001$. Details of the statistical analyses are presented in Supplementary Table 1. See also new Supplementary Figs. 5e, 5f, 11a, 11b.

3. Similarly, experiments using a within subjects design to assess the effects of SNI, particularly on evoked calcium transient activity, should include sham control groups or otherwise a measure that demonstrates the stability of emitted fluorescence at baseline or an alternative task where SNI-induced effects have been demonstrated through other experiments to be unaffected. Otherwise the potential loss of signal

over time cannot be ruled out.

RESPONSE: We'd like to thank the reviewer for this valuable suggestion. GCaMP6 imaging has been widely used in neuroscience to evaluate transient changes of calcium signals, but without sham control groups, it is difficult to tell whether the decrease of GCaMP6 signals reflect a transient down-regulation of neural activity or a gradual decay of signals over time.

As suggested, we now included sham control groups in all fiber photometry experiments. We observed no overt changes in both von-Frey filament and sucrose licking-evoked calcium transient activity at 2 weeks and 6 weeks after surgery, respectively (Response Fig. 3, also see new Supplementary Fig. 3d, 3e, 9d, 9e and 14). These data demonstrated that the altered calcium activity in SNI mice was not due to loss of signals over time.

We have incorporated these data to the revised manuscript.

Response Fig 3. Summarized data of Ca²⁺ and DA_{2m} signals in sham mice. a, Schematic of the experimental design. **b, d, g, j,** Averaged responses (left), heatmaps (middle) and AUC during 0-5 s (right) showing fluorescence responses evoked by 0.4g von Frey filament in pre- and post-Sham 2W mice. **c, e, h, k,** Averaged responses (left), heatmaps (middle) and AUC during 0-5 s (right) showing fluorescence responses evoked by sucrose licking in pre- and post-Sham 6W mice. **f, i,** Averaged

responses (left), heatmaps (middle) and AUC during 0-5 s (right) showing DA2m signals evoked by optogenetic activation of VGluT3^{DRN→VTA} terminals in pre- and post-Sham 2W mice. Significance was assessed by two-tailed paired Student's *t*-test in (b-k). Not significant (ns). See also new Supplementary Figs. 3d, 3e, 9d, 9e and 14.

4. In addition methodological details outlining how laser stimulations in the RTPT and photometry signals were time locked to Von Frey filament stimulations and bouts of licking would aid in the interpretation of the findings.

RESPONSE: Laser stimulations were used in the CPP/CPA test, but not in the RTPT. We apologize for any confusion caused by the lack of clarity. As suggested, we have added the following details in method part as below:

“To assess pain sensitivity following optostimulation, mice were given high frequency tonic (10 ms duration at 20 Hz) blue (473 nm, 3-5 mW) or yellow (594 nm, 8-10 mW) light stimulation 2 min before von Frey filament or thermal stimuli. For ChR2 activation during the sucrose preference test, tonic blue light stimulation (30 min on/30 min off/30 min on/30 min off) was applied during test day. For CPP/CPA assay, 30 min-tonic blue/yellow light stimulation was applied during training day.” (please see page 59, lines 1168-1174 in the revised manuscript)

“For fiber photometry recordings in the von Frey test, the filament stimulation (0.4 g) was delivered onto the ipsilateral hind paw (nerve-injured or sham operation side) for 2 s six times after 30 min habituation, unless that there was a withdrawal of the paw, stimulation was terminated at any moment. To avoid sensitization to the stimuli, inter-trial intervals were designed as approximately 120 s. Ca²⁺ responses evoked by von Frey stimulation in SNI/Sham mice compared with pre-SNI/pre-Sham mice were recorded. For the sucrose licking test, mice were habituated with a bottle of 2% sucrose for 24 h followed by water deprivation for 24 h, then were given free access to the bottle of 2% sucrose, Ca²⁺ responses evoked by sucrose licking

in SNI/Sham mice compared with pre-SNI/pre-Sham mice were recorded, and fluorescence signals during the first six times of licking behavior per mouse were analyzed. For the DA release test, tonic optostimulation (473 nm, 5 mW, 10 ms duration at 20 Hz) was delivered for 2 s through the fiber implanted in VTA. The fluorescence signals during the first six times were recorded to measure dopamine release in SNI/Sham mice compared with pre-SNI/pre-Sham mice. The onset of each event was tagged by triggering mark key (ThinkerTech), which was time-locked to the fiber photometry system, thus behaviors and fluorescence signals were captured simultaneously.” (please see page 60, lines 1204-1220 in the revised manuscript)

5. Finally, please also indicate whether sex was accounted for in the experiments – though there is mention that experiments were carried out in both sexes it should be clarified if efforts were made to include an equal representation within experiments. Collectively, the studies capture a novel mechanism involved in chronic pain which are expected to advance the field of pain research.

RESPONSE: We thank this reviewer for raising the issue of potential differences related to gender, which is extremely important in the pain field. In our experiments, especially in all pain assessment experiments, we made efforts to include an equal representation. Our results demonstrated that chemogenetic activation of DA^{VTA} neurons (Response Fig. 4 a, b, also see new Supplementary Fig. 5c, d) and optogenetic activation of VGluT3^{DRN→VTA} terminals (Response Fig. 4c, d, also see new Fig. 2c, d, new Supplementary Fig. 11) relieved the mechanical and thermal hypersensitivity of either sex to the similar extent. Consistent with this, we observed no sex difference in VGluT3^{DRN→VTA} terminals inhibition-induced hypersensitivity (Response Fig. 4c, d, also see new Fig. 3d, e). In the revised version, we pooled data from different sexes together in most of our experiments (see below).

Response Fig 4. Pain modulation by the DA^{VTA} neurons and VGlut3^{DRN→VTA} terminals in both sexes. a, b, Mechanical paw withdrawal threshold (a) and thermal paw withdrawal latency (b) of hM3Dq-mCherry-expressing and mCherry-expressing *DAT-Cre* mice with saline or CNO treatment. c, d, Mechanical paw withdrawal threshold (c) and thermal paw withdrawal latency (d) of ChR2-mCherry-expressing and mCherry-expressing Sham or SNI mice with (on) or without (off) optogenetic stimulation. e, f, Mechanical paw withdrawal threshold (e) and thermal paw withdrawal latency (f) of eNpHR-EYFP-expressing and EYFP-expressing mice with (on) or without (off) optogenetic stimulation. For all panels, the sex of individual data point was labeled. Significance was assessed by two-way ANOVA followed by Bonferroni's multiple comparisons test. All data are presented as the mean \pm s.e.m. **P*

< 0.05, ** $P < 0.01$, *** $P < 0.001$, not significant (ns). See also new Supplementary Fig. 5c, d and Figs. 2c, 2d, 3d, 3e.

Reviewer #2 (Remarks to the Author):

Uncovering the neuronal circuitry mediating pain may provide opportunities to develop new therapies for pain treatment and limit the development of comorbid disorders. Here, the authors used multiple techniques, including fiber photometry, electrophysiology, optogenetics, chemogenetics, and rabies virus tracing, to investigate the involvement of Vglut3DRN-DAVTA -NacMed circuit in the regulation of chronic pain and comorbid depression. However, I have major concerns about several places.

RESPONSE: We thank this referee for his/her insightful critiques and suggestions which provided us great opportunities to strengthen our conclusions.

Major concerns:

1. Several conclusions and figures are very similar to previous studies. For example,
 - a) The conclusion “VTA neurons from SNI had lower firing rates” in Fig. 1 has been reported in Ren et al., 2016.
 - b) The conclusion “most DR neurons projecting to the VTA express VGluT3” in Fig. 2 has been reported in Qi et al., 2014. Rabies tracing of dopamine neurons in the VTA has been reported by multiple groups as well.

Repeated work in comparison with previous studies also presented in several supplementary figures. For example, findings on “inflammatory pain decreased the activity of ventral tegmental area (VTA) dopamine (DA) neurons” and “the decreased activity of DA neurons was associated with reduced motivation for natural rewards, consistent with anhedonia-like behavior” has been reported in details in Markovic et al., 2021.

I strongly suggest the authors compress the paper into four figures, and only focus on the actual novel findings. And please cite and discuss relevant work thoroughly. Two-bottle test is a standard assay for anhedonia-like behavior, without other behavioral assays supporting the conclusion on comorbid depressive-like behavior

(such as learned helpless, forced swimming, and etc.), please just say anhedonia-like behavior. Using a different phrasing is not going to change the fact about the novelty of the work, but only reduces accuracy.

RESPONSE: We appreciate a lot for these constructive suggestions to improve our manuscript. As recommended, we removed all data that were inconsistent with those that have been reported into supplementary figures, and compressed our novel findings into four main figures (please see new Figures, and also new Supplementary Figures).

In addition, we changed “depressive-like behavior” into “anhedonia-like behavior” throughout the manuscript.

Meanwhile, we have cited and discussed relevant work in the results and discussion sections. As shown below, in the discussion part:

“For instance, both rodents and clinical studies reported that chronic pain induces hypodopaminergic tone^{6,8,9,37,38}, resulting in anhedonia and depression^{2,9,39,40}.” (please see page 16, lines 448-450 in the revised manuscript)

“In addition to the reduced firing frequency of DA^{VTA} neurons in neuropathic pain (Supplementary Fig. 4c), which was reported by several groups^{8,37,40}” (please see page 16, lines 454-455 in the revised manuscript)

“Our viral–genetic tracing and electrophysiological data demonstrate that VGluT3⁺ neurons in the DRN mainly connect with DA^{VTA} neurons (Supplementary Fig. 6, 7), consistent with a previous report¹⁵. Our data further show that DRN VGluT3⁺ neurons make both glutamatergic and serotonergic synapses with DA^{VTA} neurons (Supplementary Fig. 8), which is in line with previous findings showing that a subset of DRN neurons project to VTA co-release glutamate and serotonin^{15,30,48}.” (please see page 17, lines 475-479 in the revised manuscript)

2. The authors stated that “our findings reasoned a causal connection between the

pathophysiological decreases in I_h with stage-dependent CBD manifestation during neuropathic pain”(line 429). However the data presented in this study is only supporting a correlation between the two subjects but not causal relationships.

A causal relationship between “pathophysiological decreases in I_h with stage-dependent CBD manifestation during neuropathic pain” can only be established if gain-of-function and loss-of-function studies have been done with I_h and the consequences are analysed.

RESPONSE: We fully agree that the statement “*our findings reasoned a causal connection between the pathophysiological decreases in I_h with stage-dependent CBD manifestation during neuropathic pain*” (line 429) lacks experimental evidence. In the revised version, we changed it to “*our findings suggest potential correlations between the pathophysiological decreases in I_h and manifestation of stage-dependent CAB during neuropathic pain. Further functional manipulations are required to test such possibilities.*” (please see page 17, lines 462-465 in the revised manuscript).

3. As mentioned in the manuscript that Vglut3DRN-DAVTA pathway promotes reward, it’s not surprising that inhibiting this pathway causes CPA. Thus, it’s not good evidence to link this pathway with ongoing pain.

CPA is not “an effective assay for measuring ongoing pain” (line 321). It never is. Thus conclusion of “prolonged silencing of Vglut3DRN neurons mimic both chronic pain-like hypersensitivity and CDB” (line 342) is unfounded.

RESPONSE: We thank this reviewer for pointing out our incorrect statement “CPA, an effective assay for measuring ongoing pain” (line 321). We corrected the statement as “We further assessed whether an aversive affective memory could be evoked using the conventional conditioned place aversion (CPA) behavioral assay.”

Given that the $VGluT3^{DRN} \rightarrow DA^{VTA}$ circuit promotes reward, we completely agree with this reviewer that ‘inhibiting a reward pathway itself causes CPA’ is not surprising and therefore it is questionable to use CPA as an indicator of reduction of pain perception. In the revised version, we changed the conclusion “prolonged silencing of $VGluT3^{DRN}$ neurons mimic both chronic pain-like hypersensitivity and

CDB” to “prolonged silencing of VGluT3^{DRN} neurons mimic both chronic pain-like reflexive hypersensitivity and CAB”, since reflexive hypersensitivity was supported by the von-Frey test. Meanwhile, to avoid potential confusion of CPA results and also to present the data according to the experimental timeline (sequentially performed CPA, von Frey test and SPT), we changed the order of Fig 5i and 5K in the original manuscript.

We also include discussion related to the CPP/CPA measurements in the revised manuscript:

“Pain has both sensory and aversive dimensions. In addition to assessing its sensory component with external stimuli-evoked reflex responses, we used the CPP assay to measure whether the reduction of an aversive state (pain relief) could be achieved following VGluT3^{DRN}→DA^{VTA} circuit excitation, and used the CPA assay to assess whether the aversive state could be induced following VGluT3^{DRN}→DA^{VTA} circuit inhibition. Given the role of VGluT3^{DRN}→VTA projection in mediating reward^{15,20}, the circuit manipulation itself could cause CPP/CPA. It is thus difficult to conclude that the results of the CPP and CPA experiments reflect changes of pain affection in our study. However, our evidence establishes a compelling correlation between aberrant VGluT3^{DRN}→DA^{VTA} circuit activity and SNI-induced sensory hypersensitivity and CAB.” (please see page 18, lines 488-498 in the revised manuscript).

4. If the authors want to establish the pathway of Vglut3DRN->DAVTA->Medial Nac, they need to use cTRIO(Schwarz et al. 2015), to show that DAVTA neurons projecting to Medial Nac indeed receive input from Vglut3+ DRN neurons. And because Vglut3+ DRN neurons also directly project to Nac, the authors need to show the Vglut3DRN->DAVTA->Medial Nac pathway is the predominant one for the findings but not Vglut3DRN-> Nac.

RESPONSE: We thank this reviewer for this valuable advice. As suggested, we used

the cTRIO system to investigate the $VGluT3^{DRN} \rightarrow DA^{VTA} \rightarrow NAcMed$ pathway. To achieve this, we injected AAV-retro-DIO-Flp into the NacMed, and AAV-fDIO-TVA-GFP/ AAV-fDIO-RVG into the VTA of *DAT-Cre* mice. Two weeks later, RV-EnA- Δ G-DsRed was injected into the VTA. Our results showed that DsRed-labeled neurons were detected in many brain regions including the DRN (Response Fig. 5, also see new Supplementary Fig. 13). In addition, ~68% of DsRed-labeled DRN neurons expressed VGluT3, suggesting that NAcMed-projecting DA^{VTA} neurons receive input from $VGluT3^{DRN}$ neurons.

To address the concern about whether the $Vglut3^{DRN} \rightarrow DA^{VTA} \rightarrow NAcMed$ pathway is the predominant one, we examined whether $VGluT3^{DRN} \rightarrow NAcMed$ pathway also contributes to the pain and anhedonia relief. We injected *VGluT3-Cre* mice with AAV-DIO-ChR2-mCherry or AAV-DIO-mCherry into the DRN and implanted optical fibers above the NAcMed (Response Fig. 6a, b, also see new Supplementary Fig. 16a, b). We found that optogenetic activation of the $VGluT3^{DRN}$ neural terminals within the NAcMed has no effects on reducing both mechanical and thermal hypersensitivity in post-SNI 2W and 6W mice (Response Fig. 6c-f, also see new Supplementary Fig. 16c-f). The comorbid anhedonia-like behavior was also unaffected (Response Fig. 6g, also see new Supplementary Fig. 16g). Thus, we propose that the $VGluT3^{DRN} \rightarrow DA^{VTA} \rightarrow NAcMed$ rather than $VGluT3^{DRN} \rightarrow NAcMed$ pathway plays a predominant role in relieving chronic pain and CAB.

We have incorporated these data to the revised manuscript.

Response Fig 5. Presynaptic input to NAcMed-projecting DA^{VTA} neurons revealed by rabies-mediated trans-synaptic tracing. **a**, Schematic of cTRIO based retrograde monosynaptic tracing using *DAT-Cre* mice. **b**, Representative images of the starter cells in the VTA (left) and RV-DsRed-labeled cells in the DRN (middle) which co-localize with VGlut3 immunofluorescence (right). Starter cells (yellow) co-expressing AAV-fDIO-TVA-GFP, AAV-fDIO-RVG (green), and rabies RV-EnvA-ΔG-DsRed (red). Scale bars, 50 μm (upper) and 200 μm (bottom). **c**, Percentage of DsRed-labeled neurons that express VGlut3 in *DAT-Cre* mice, n = 9 sections from three mice. **d**, Representative images showing DsRed-expressing cells (red) that make monosynaptic contact onto NAcMed-projecting DA^{VTA} neurons. Scale bars, 200 μm. PBN, parabrachial nucleus; LC, locus coeruleus; LDTg, laterodorsal tegmentum; PAG, periaqueductal gray; DRN, dorsal raphe nucleus; SNR, substantia nigra pars reticulata; LH, lateral hypothalamus; LHb, lateral habenular nucleus; MHb, medial habenular nucleus; CeA, central nucleus of the amygdala; BNST, bed nucleus

of the stria terminalis; ACC, anterior cingulate cortex; NAc, nucleus accumbens; mPFC, medial prefrontal cortex. Data in (c) are shown as box and whisker plots (medians, quartiles (boxes), and ranges minimum to maximum (whiskers)). **See also new Supplementary Fig. 13.**

Response Fig 6. Effects of activation of $VGluT3^{DRN}$ neural terminals within the NAcMed on chronic pain hypersensitivity and CAB. a, Schematic of the experimental design. **b,** Schematic of DRN injection of AAV-DIO-ChR2-mCherry/AAV-DIO-mCherry and representative images showing NAcMed optical fiber implantation in *VGluT3-Cre* mice. Scale bars, 500 μ m. **c-f,** Mechanical paw withdrawal threshold (**c, e**) and thermal paw withdrawal latency (**d, f**) of ChR2-mCherry-expressing and mCherry-expressing SNI mice with (on) or without

(off) optogenetic stimulation. **g**, Preference for sucrose in the SPT. Significance was assessed by two-way ANOVA followed by Bonferroni's multiple comparisons test in **(c-f)** and two-tailed unpaired Student's *t*-test in **(g)**. All data are presented as the mean \pm s.e.m. not significant (ns). Details of the statistical analyses are presented in Supplementary Table 1. **See also new Supplementary Fig. 16.**

5. There are also some inconsistencies between parts of the results in this manuscript and previous publications. Please provide more information.

a. Whether VTA DA neurons are activated by aversive stimuli. Ungless et al., 2004 reported that VTA dopamine neurons are specifically excited by reward, while a population of nondopamine neurons in the VTA is excited by aversive stimuli. Markovic et al., 2021 also reported that pain suppresses VTA DA neuron activity. de Jong et al., 2019 suggested a subpopulation, vNAcMed-projecting VTA DA neurons are activated by the initial foot shock. However, that experiment was conducted by terminal recording rather than soma recording.

RESPONSE: We thank this reviewer for pointing out discrepancies in how of DA neurons respond to aversive stimuli.

As mentioned by this reviewer, Ungless (2004) reported that a group of non-DA neurons were activated by aversive stimuli in the VTA. However, in a follow-up study, Ungless's lab reported functional diversity of DA neurons in the VTA: one group of DA neurons in the ventral VTA are phasically excited by foot shocks, while another group of DA neurons located in the dorsal VTA are inhibited by noxious foot shocks and display an excitation at the termination of the stimulus (Brischoux et al., 2009). These observations suggested DA^{VTA} neurons exhibit diverse response to reward or aversive stimuli. To further investigate this, we delivered unexpected air puffs, which was classically viewed as an aversive stimulation (Moriarty et al., 2012), to the eye of mice and found that DA^{VTA} neurons and VGlut3^{DRN→VTA} afferents were both activated by air puffs in naïve mice (**Response Fig. 7**).

In addition, how DA^{VTA} neurons respond to aversive stimuli might depend on different conditions (e.g. physiology vs pathology). For example, in normal conditions,

relief of pain activates the mesolimbic dopamine circuit to facilitate learning and promote behavior (Baliki et al., 2010; Navratilova et al., 2015). Under chronic pain condition, the DA^{VTA} neuron activity was inhibited as pain challenge (Martikainen et al., 2015). In our study, 0.4g von Frey filament which is an aversive pain stimulus in SNI mice caused VTA DA neural inhibition and reduced DA release in NAcMed (new Fig. 1d and Fig. 4e), in consistent with results reported in Markovic et al., 2019, Ren et al., 2016.

In the revised version, we discuss the inconsistency of the response of DA^{VTA} neurons to aversive stimuli:

“The role of DA^{VTA} neurons in reward processing has long been recognized. However, their roles in encoding aversive information including pain, remains controversial. Several studies showed that a large proportion of DA^{VTA} neurons are inhibited by acute noxious stimuli^{11,35}, whereas others demonstrated that a subpopulation of DA^{VTA} neurons are excited by acute pain stimuli^{19,20}. The discrepancy could arise from the anatomical and functional heterogeneity of DA^{VTA} neurons^{15,19,20,36}. For example, DA neurons in the ventral VTA are phasically excited by footshocks, whereas those located in the dorsal VTA are inhibited by the same stimulus¹⁹. In addition, different conditions (physiology vs pathology) could also affect the dopaminergic tone. For instance, both rodents and clinical studies reported that chronic pain induces hypodopaminergic tone^{6,8,9,37,38}, resulting in anhedonia and depression^{2,9,39,40}.” (please see page 16, lines 439-450 in the revised manuscript)

Response Fig 7. Ca^{2+} activity in DA^{VTA} neurons in response to aversive stimuli in naïve mice. **a, d** Schematic of the experimental design. **b, c, e, f**, Averaged responses (**b, e**) and heatmaps (**c, f**) showing Ca^{2+} responses evoked by air puff stimulation in naïve mice.

b. Whether the medial Nac is the target of the Vglut3DRN-DAVTA pathway. de Jong et al., 2019 reported a different statement that DR VGLUT3 Inputs to VTA activate NAcLat-Projecting DA neurons.

RESPONSE: We appreciated a lot for this reviewer to raise this point.

In our study, we observed DRN-targeted VTA neurons project to both NAcMed and NAcLat (Please see new Supplementary Fig. 5), releasing DA in both regions following $\text{VGlut3}^{\text{DRN} \rightarrow \text{VTA}}$ terminals activation (Please see new Figure 4). These results suggest that $\text{VGlut3}^{\text{DRN}}$ inputs to VTA activate DA neurons projecting to both NAcMed and NAcLat. Anatomically, our study is very consistent with that reported by de Jong et al., 2019, both of which revealed projections from VTA DA neurons to the NAcMed and NAcLat.

The discrepancy between two studies lies on the functional dominance of $\text{VTA}^{\text{DA}} \rightarrow \text{NAcMed}$ vs NAcLat in mediating reward. By using *in vitro* electrophysiological recordings, de Jong et al. reported that activating $\text{VGlut3}^{\text{DRN} \rightarrow \text{VTA}}$ terminals evoked more robust EPSCs (larger amplitude and frequencies) in NAcLat-projecting vs ventral NAcMed-projecting DA neurons (the immediate downstream target of $\text{VGlut3}^{\text{DRN}}$ in the VTA). In our case, we performed

in vivo fiber photometry to record DA release (the final outcome) both in the NAcMed and NAcLat when VGluT3^{DRN→VTA} terminals were opto-stimulated.

It is noteworthy that the VTA contains heterogeneous cell populations including dopaminergic, glutamatergic, and GABAergic neurons, many of which co-release distinct neurotransmitters (Morales et al., 2017; Hnasko et al., 2010). Optostimulating VGluT3^{DRN→VTA} terminals lacks of specificity on activating DA^{VTA} neurons. It is therefore likely that the activation of non-dopaminergic neurons contribute to dopamine release in the NAc region in our case. Therefore, the discrepancies between these two studies are likely due to different methodology and neural targets.

We also include discussion related to this point in the revised manuscript:

“In addition, there has been controversy regarding the VGluT3^{DRN} inputs to NAc-projecting DA^{VTA} neurons. A previous study found that activating VGluT3^{DRN} terminals produced larger EPSCs with more frequencies in NAcLat-projecting rather than ventral NAcMed-projecting DA neurons, suggesting that the NAcLat-projecting DA neurons are predominant in promoting reward²⁰. In our study, by using fiber photometry, we observed that opto-stimulating VGluT3^{DRN→VTA} terminals caused DA release both in the NAcMed and NAcLat. Interestingly, neuropathic pain evoked region-specific DA alterations preferentially occur in the NAcMed (Fig. 4). Such discrepancy might result from the different methodology and targeted populations for recording. We speculate that the VGluT3^{DRN→DA^{VTA}→NAcMed} circuit proposed here largely participates in the chronic pain-induced decrease of dopamine release, but not in aversion-related excitation.” (please see page 19, lines 521-533 in the revised manuscript).

c. Whether VTA-Nac pathway regulates hyperalgesia. In the study of Markovic et al., 2021, they reported that activation of VTA–NAc-projecting DA neurons did not alter hyperalgesia induced by complete Freund’s adjuvant (CFA) injection.

RESPONSE: We thank the reviewer for raising this point. Markovic et al., 2021

reported that activation of the VTA→NAc pathway is sufficient to reverse the pain-induced decrease in motivated behavior without affecting the sensory component of inflammatory pain. By contrast, Yang et al., 2021 and Sato et al., 2022 both showed that activation of DA^{VTA}→NAc pathway does modulate the sensory component of pain, and thus providing support of our findings. These studies together with our results indicate that distinct manipulation of the mesolimbic system may differentially contribute to pain hypersensitivity.

Several possible contributing factors could have led to these contradictory results.

- 1) The input-output of midbrain DA neurons are highly heterogeneous (Beier et al., 2015). Besides the VGluT3^{DRN}, the NAcMed-projecting DA^{VTA} neurons also receive inputs from brain regions including the LDTg, LC, PAG, LHB (Supplementary Fig. 13d). Since distinct inputs to the NAcMed-projecting DA^{VTA} neurons could affect sensory and motivational behaviors respectively (Yang et al., 2021; de Jong et al., 2019), it is likely that they impact different components of pain in an input dependent manner.
- 2) Different animal models and time points for measuring pain could result in distinct results. Previous reports have demonstrated that dysfunction of the mesolimbic DA reward circuit develops in a time-dependent manner (Kato et al., 2016). Markovic et al., 2021 assessed thermal hyperalgesia using a different pain hypersensitivity model (CFA vs SNI) at a much earlier stage (48-h post CFA vs 2W/6W post SNI).

We also include discussion related to this point in the revised manuscript:

“A previous study found that activation of the VTA→NAc pathway is sufficient to reverse the pain-induced decrease in motivated behavior without affecting the sensory component of inflammatory pain⁹. We found that the VGluT3^{DRN}→DA^{VTA} circuit modulates reflective hypersensitivity and CAB via DA release in NAcMed. Thus, different manipulations of the mesolimbic system could result in distinct alterations of pain behavior. One plausible explanation could be due to the input-output heterogeneity of midbrain DA neuron^{27,57}. In light of this, our data showed that, besides the VGluT3^{DRN}, the

NAcMed-projecting DA^{VTA} neurons receive inputs from other brain regions including the LDTg, LC, PAG, Lhb (Supplementary Fig. 13d). Since distinct inputs to the NAcMed-projecting DA^{VTA} neurons could affect sensory and motivational behaviors^{11,20,22,38,54}, it is likely that they impact different components of pain in an input dependent manner. Additionally, varying time points and injury models for measuring pain could also result in different conclusions⁵⁸.” (please see pages 19-20, lines 544-556 in the revised manuscript).

Mirror concerns:

1. Would it be possible that the decreased calcium responses (especially the response to sucrose drinking in SNI 6W) in Fig. 1 and 3 are caused by signal decay?

RESPONSE: We thank the reviewer for raising this concern, which was also raised by Reviewer1. To address this, we included sham control groups in all fiber photometry experiments. We observed no overt changes in both von-Frey filament and sucrose licking-evoked calcium transient activity at 2 weeks and 6 weeks after surgery, respectively (Response Fig. 3, also see new Supplementary Fig. 3d, 3e, 9d, 9e and 14). These data demonstrated that the altered calcium activity in SNI mice was not due to loss of signals over time.

We have incorporated these data to the revised manuscript.

Response Fig 3. Summarized data of Ca²⁺ and DA_{2m} signals in sham mice. a, Schematic of the experimental design. **b, d, g, j,** Averaged responses (left), heatmaps (middle) and AUC during 0-5 s (right) showing fluorescence responses evoked by 0.4g von Frey filament in pre and post-Sham 2W mice. **c, e, h, k,** Averaged responses (left), heatmaps (middle) and AUC during 0-5 s (right) showing fluorescence responses evoked by sucrose licking in pre- and post-Sham 6W mice. **f, i,** Averaged

responses (left), heatmaps (middle) and AUC during 0-5 s (right) showing DA2m signals evoked by optogenetic activation of VGluT3^{DRN→VTA} terminals in pre- and post-Sham 2W mice. Significance was assessed by two-tailed paired Student's *t*-test in (b-k). Not significant (ns). **See also new Supplementary Figs. 3d, 3e, 9d, 9e and 14.**

2. Please add asterisk (between SNI 2W and SNI 6W) in Fig. 1j to support the statement in Page 5 of the manuscript that “dramatic *I_h* reduction in DAT neurons from SNI 6W mice ...”

RESPONSE: As suggested, we added asterisk (between SNI 2W and SNI 6W) in Fig. 1j.

3. Please provide more detailed method of optostimulation in Fig. 4c (phasic or tonic? Coupled with filament stimulation or not?)

RESPONSE: We appreciate this reviewer's suggestion. In Methods section, we have added details as below:

“To assess the pain sensitivity following optostimulation, mice were given high frequency tonic (10 ms duration at 20 Hz) blue (473 nm, 3-5 mW) or yellow (594 nm, 8-10 mW) light stimulation 2 min before von Frey filament or thermal stimuli.” (please see page 59, lines 1168-1171 in the revised manuscript).

4. Fig. 5i, prolonged hypersensitivity was observed 6W after CNO injection. Curious whether any neural adaption (e.g., dampened activity of Vglut3DRN-DAVTA pathway) also can be seen long after the termination of CNO injection.

RESPONSE: We thank this reviewer for raising this insightful question. To address this, we injected AAV-DIO-hM4Di-mCherry/AAV-DIO-mCherry and AAV-DIO-ChR2 simultaneously into the DRN of *VGluT3-Cre* mice. Six weeks after CNO injection, we recorded light-evoked EPSC, firing rate, and *I_h* of DA^{VTA} neurons. Intriguingly, amplitudes of light-evoked EPSCs of DA^{VTA} neurons in brain slices from

hM4Di-expressing post-CNO 6W mice were significantly decreased when compared with the control post-CNO 6W mice (Response Fig. 8, also see new Fig. 3m). Firing frequency and I_h were also reduced (Response Fig. 8, also see new Fig. 3n, o). These data suggest that the $VGluT3^{DRN} \rightarrow DA^{VTA}$ circuit underlies adaptation long after silencing $VGluT3^{DRN}$ neurons, which is consistent with the plastic changes observed in post-SNI 6W mice (new Fig. 1m, s, v).

We have added these data to the revised manuscript.

Response Fig 8. Synaptic adaption of the $VGluT3^{DRN} \rightarrow DA^{VTA}$ pathway after CNO injection. **a**, Schematic of the experimental design. **b**, Light-evoked EPSCs recorded from $VGluT3^{DRN}$ -targeted DA^{VTA} neurons of hM4Di-mCherry-expressing or mCherry-expressing mice. **c**, **d**, Action potential firing rate (**c**) and I_h at -120 mV (**d**) recorded from $VGluT3^{DRN}$ -targeted postsynaptic DA^{VTA} neurons. Significance was assessed by two-way ANOVA followed by Bonferroni's multiple comparisons test in (**c**) and Mann-Whitney U test in (**b**, **d**). * $P < 0.05$, ** $P < 0.01$, not significant (ns). Details of the statistical analyses are presented in Supplementary Table 1.

5. What is the reference of line 48-51?

RESPONSE: We apologize for omitting the reference and have now added them. (please see page 2, lines 49-51 in the revised manuscript).

6. Please change the title of Figure 2 with a more specific one.

RESPONSE: We have changed the title as "Dopamine neurons are the primary postsynaptic target of the $VGluT3^{DRN}$ neurons" and changed Figure 2 as new Supplementary Figure 8 in the revised manuscript.

7. What are the references of line 442-443, “the functional glutamatergic and serotonergic synapse ... by electrophysiology”?

RESPONSE: The conclusion that “the functional glutamatergic and serotonergic synapse ... by electrophysiology” is drawn from our data. We apologize for any confusion here. We have now revised it as “And the DRN VGluT3⁺ neurons make both glutamatergic and serotonergic synapses with DA^{VTA} neurons (Supplementary Fig. 8)”.

8. Please find a native speaker to help with the writing.

RESPONSE: Thanks for the advice. We have polished the manuscript with the help of a native speaker.

Reference:

de Jong, J. W., Afjei, S. A., Pollak Dorocic, I., Peck, J. R., Liu, C., Kim, C. K., Tian, L., Deisseroth, K., & Lammel, S. (2019). A Neural Circuit Mechanism for Encoding Aversive Stimuli in the Mesolimbic Dopamine System. *Neuron*, 101(1), 133-151.e7. <https://doi.org/10.1016/j.neuron.2018.11.005>

Markovic, T., Pedersen, C. E., Massaly, N., Vachez, Y. M., Ruyle, B., Murphy, C. A., Abiraman, K., Shin, J. H., Garcia, J. J., Yoon, H. J., Alvarez, V. A., Bruchas, M. R., Creed, M. C., & Morón, J. A. (2021). Pain induces adaptations in ventral tegmental area dopamine neurons to drive anhedonia-like behavior. *Nature Neuroscience*, 24(11), 1601–1613. <https://doi.org/10.1038/s41593-021-00924-3>

Qi, J., Zhang, S., Wang, H. L., Wang, H., de Jesus Aceves Buendía, J., Hoffman, A. F., Lupica, C. R., Seal, R. P., & Morales, M. (2014). A glutamatergic reward input from the dorsal raphe to ventral tegmental area dopamine neurons. *Nature Communications*, 5. <https://doi.org/10.1038/ncomms6390>

Ren, W., Centeno, M. V., Berger, S., Wu, Y., Na, X., Liu, X., Kondapalli, J., Apkarian, A. V., Martina, M., & Surmeier, D. J. (2016). The indirect pathway of the nucleus accumbens shell amplifies neuropathic pain. *Nature Neuroscience*, 19(2), 220–222.

<https://doi.org/10.1038/nrn.4199>

Schwarz LA, Miyamichi K, Gao XJ, Beier KT, Weissbourd B, DeLoach KE, Ren J, Ibanes S, Malenka RC, Kremer EJ, Luo L. Viral-genetic tracing of the input–output organization of a central noradrenaline circuit. *Nature*. 2015 Aug 6;524(7563):88-92.

Ungless, M. A., Magill, P. J., & Bolam, J. P. (2004). Uniform Inhibition of Dopamine Neurons in the Ventral Tegmental Area by Aversive Stimuli. *Science*, 303(5666), 2040–2042. <https://doi.org/10.1126/science.1093360>

Reference:

Brischoux, F., Chakraborty, S., Brierley, D. I., & Ungless, M. A. (2009). Phasic excitation of dopamine neurons in ventral VTA by noxious stimuli. *Proceedings of the National Academy of Sciences of the United States of America*, 106(12), 4894–4899. <https://doi.org/10.1073/pnas.0811507106>

Moriarty, O., Roche, M., McGuire, B. E., & Finn, D. P. (2012). Validation of an air-puff passive-avoidance paradigm for assessment of aversive learning and memory in rat models of chronic pain. *Journal of neuroscience methods*, 204(1), 1–8. <https://doi.org/10.1016/j.jneumeth.2011.10.024>

Baliki, M. N., Geha, P. Y., Fields, H. L., & Apkarian, A. V. (2010). Predicting value of pain and analgesia: nucleus accumbens response to noxious stimuli changes in the presence of chronic pain. *Neuron*, 66(1), 149–160. <https://doi.org/10.1016/j.neuron.2010.03.002>

Navratilova, E., Atcherley, C. W., & Porreca, F. (2015). Brain Circuits Encoding Reward from Pain Relief. *Trends in neurosciences* 38(11), 741-750. <https://doi.org/10.1016/j.tins.2015.09.003>

Martikainen, I. K., Nuechterlein, E. B., Peciña, M., Love, T. M., Cummiford, C. M., Green, C. R., Stohler, C. S., & Zubieta, J. K. (2015). Chronic Back Pain Is Associated with Alterations in Dopamine Neurotransmission in the Ventral Striatum. *The Journal of neuroscience : the official journal of the Society for Neuroscience*, 35(27), 9957–9965. <https://doi.org/10.1523/JNEUROSCI.4605-14.2015>

Morales, M., & Margolis, E. B. (2017). Ventral tegmental area: cellular heterogeneity, connectivity and behaviour. *Nature reviews. Neuroscience*, 18(2), 73–85. <https://doi.org/10.1038/nrn.2016.165>

Hnasko, T. S., Chuhma, N., Zhang, H., Goh, G. Y., Sulzer, D., Palmiter, R. D., Rayport, S., & Edwards, R. H. (2010). Vesicular glutamate transport promotes dopamine storage and glutamate corelease in vivo. *Neuron*, 65(5), 643–656. <https://doi.org/10.1016/j.neuron.2010.02.012>

Yang, H., de Jong, J. W., Cerniauskas, I., Peck, J. R., Lim, B. K., Gong, H., Fields, H. L., & Lammel, S. (2021). Pain modulates dopamine neurons via a spinal-parabrachial-mesencephalic circuit. *Nature neuroscience*, 24(10), 1402–1413. <https://doi.org/10.1038/s41593-021-00903-8>

Sato, D., Narita, M., Hamada, Y., Mori, T., Tanaka, K., Tamura, H., Yamanaka, A., Matsui, R., Watanabe, D., Suda, Y., Senba, E., Watanabe, M., Navratilova, E., Porreca, F., Kuzumaki, N., & Narita, M. (2022). Relief of neuropathic pain by cell-specific manipulation of nucleus accumbens dopamine D1- and D2-receptor-expressing neurons. *Molecular brain*, 15(1), 10. <https://doi.org/10.1186/s13041-021-00896-2>

Beier, K. T., Steinberg, E. E., DeLoach, K. E., Xie, S., Miyamichi, K., Schwarz, L., Gao, X. J., Kremer, E. J., Malenka, R. C., & Luo, L. (2015). Circuit Architecture of VTA Dopamine Neurons Revealed by Systematic Input-Output Mapping. *Cell*, 162(3), 622–634. <https://doi.org/10.1016/j.cell.2015.07.015>

Kato, T., Ide, S., & Minami, M. (2016). Pain relief induces dopamine release in the rat nucleus accumbens during the early but not late phase of neuropathic pain. *Neuroscience letters*, 629, 73–78. <https://doi.org/10.1016/j.neulet.2016.06.060>

REVIEWERS' COMMENTS

Reviewer #1 (Remarks to the Author):

They authors had done a great job addressing all my comments. I don't have any further comments.

Reviewer #2 (Remarks to the Author):

The authors have addressed most of my concerns.

1. For the cTRIO data in Supplementary Fig 13, please present the animal numbers involved in this experiment and the GFP channel images of DRN and report the number of GFP+ cells in DRN if there are any. Because the VTA and the DRN are quite close to each other, there could be AAV-fDIO-TVA-GFP contaminations in DRN during injection, in which case DsRed+ cells in the DRN are "contaminated starter cells" rather than the trans-synaptically labelled neurons.

2. Please present the "p-value" and "n" for all the figures that contain statistical analysis.

REVIEWER COMMENTS

Reviewer #1 (Remarks to the Author):

They authors had done a great job addressing all my comments. I don't have any further comments.

RESPONSE: We appreciate the reviewer's thoughtful comments throughout the review process.

Reviewer #2 (Remarks to the Author):

The authors have addressed most of my concerns.

1. For the cTRIO data in Supplementary Fig 13, please present the animal numbers involved in this experiment and the GFP channel images of DRN and report the number of GFP+ cells in DRN if there are any. Because the VTA and the DRN are quite close to each other, there could be AAV-fDIO-TVA-GFP contaminations in DRN during injection, in which case DsRed+ cells in the DRN are "contaminated starter cells" rather than the trans-synaptically labelled neurons.

RESPONSE: We thank this reviewer for this valuable advice. As suggested, we presented the GFP channel images of DRN and the animal numbers. We observed no GFP expression in the DRN (Response Fig. 1, also see new Supplementary Fig. 13). The data indicate that the DsRed⁺ cells in the DRN were not due to AAV-fDIO-TVA-GFP contaminations.

We have incorporated these data to the revised manuscript.

Response Fig 1. Presynaptic input to NAcMed-projecting DA^{VTA} neurons revealed by cTRIO-based mediated trans-synaptic tracing. a, Schematic of cTRIO

based retrograde monosynaptic tracing using *DAT-Cre* mice. **b**, Representative images of the starter cells in the VTA (left) and RV-DsRed-labeled cells in the DRN (middle) which co-localize with VGluT3 immunofluorescence (right). Starter cells (yellow) co-expressing AAV-fDIO-TVA-GFP, AAV-fDIO-RVG (green), and rabies RV-EnvA-ΔG-DsRed (red). Scale bars, 50 μm (upper) and 200 μm (bottom). **c**, Percentage of DsRed-labeled neurons that express VGluT3 in *DAT-Cre* mice, n = 9 sections from three mice. **d**, GFP-expressing neurons in the DRN. n = 3 mice.

2. Please present the "p-value" and "n" for all the figures that contain statistical analysis.

RESPONSE: We thank the reviewer for pointing this out. We have added the "p-value" and "n" for all the figures that contain statistical analysis.